# Long-range atmospheric transport of microplastics across the southern hemisphere

Qiqing Chen[1], Guitao Shi [1,2] ✉, Laura E. Revell [3], Jun Zhang[4,5], Chencheng Zuo[1], Danhe Wang[2], Eric C. Le Ru [6], Guangmei Wu[2] & Denise M. Mitrano [7]

Airborne microplastics (MPs) can undergo long range transport to remote regions. Yet there is a large knowledge gap regarding the occurrence and burden of MPs in the marine boundary layer, which hampers comprehensive modelling of their global atmospheric transport. In particular, the transport efficiency of MPs with different sizes and morphologies remains uncertain. Here we show a hemispheric-scale analysis of airborne MPs along a cruise path from the mid-Northern Hemisphere to Antarctica. We present the inaugural measurements of MPs concentrations over the Southern Ocean and interior Antarctica and find that MPs fibers are transported more efficiently than MPs fragments along the transect, with the transport dynamics of MPs generally similar to those of non-plastic particles. Morphology is found to be the dominant factor influencing the hemispheric transport of MPs to remote Antarctic regions. This study underlines the importance of long-range atmospheric transport in MPs cycling dynamics in the environment.

Evidence is accumulating that microplastics (MPs; plastic particles 1–5000 μm in size) are capable of long-range atmospheric transport to remote regions[1–6]. Current estimates of plastic mass in the atmosphere span over four orders of magnitude (0.013–25 Mt)[7]. However, the occurrence, concentration, and composition of airborne MPs over the oceans, including the remote Southern Ocean, remains obscure due to a paucity of observations in these regions. Understanding the spatial patterns of MPs over the marine boundary layer (MBL) is essential to accurately model MPs transport from the continents to the MBL.

Due to their size ranges, diverse composition, and irregular morphologies, it is highly probable that the migration mechanisms of airborne MPs are substantially complex. For instance, airborne micro(nano)plastics (MnPs) measured with widely available spectroscopic methods have a typical size range of 20 μm–5 mm[7], making them noticeably larger than the aerosols typically found in the atmosphere (e.g., sea salts). Nanoplastics (plastic particles <1 μm) have been identified in the atmosphere[8] but are less well understood than MPs. Long-range atmospheric transport of giant dust particles on the order of hundreds of micrometers in size has been reported;[9] however, whether the transport efficiency of MPs and non-plastic particles is comparable, and whether different MPs morphologies exhibit different migration behaviors, remains unclear. In addition, the limited knowledge on the identification of MPs sources, emission mechanisms, and the transport and resuspension processes (i.e., the influence of ocean winds, sea spray mist[10,11]) contribute towards the overall uncertainty. Furthermore, plastics can present excess electrostatic charge for considerable amounts of time[12], which is usually neglected as they are considered as electroneutral[13], which may increase this

[1]State Key Laboratory of Estuarine and Coastal Research, East China Normal University, Shanghai 200241, China. [2]Key Laboratory of Geographic Information Science (Ministry of Education), School of Geographic Sciences, East China Normal University, Shanghai 200241, China. [3]School of Physical and Chemical Sciences, University of Canterbury, Christchurch 8140, New Zealand. [4]NYU-ECNU Physics and Mathematics Research Institutes, New York University Shanghai, Shanghai 200062, China. [5]Department of Physics, New York University, New York, NY 10003, USA. [6]The MacDiarmid Institute for Advanced Materials and Nanotechnology, School of Chemical and Physical Sciences, Victoria University of Wellington, Wellington 6140, New Zealand. [7]Department of Environmental Systems Science, ETH Zurich, Zurich 8092, Switzerland. ✉e-mail: gtshi@geo.ecnu.edu.cn

complexity (i.e., change their friction coefficients[14]) in understanding the hemispheric abundance of MPs. Amongst the numerous uncertainties related to the transportation of MPs in the atmosphere, we present evidence of MPs in the marine atmosphere of the Southern Hemisphere, which will help in understanding the both roles of the ocean and long-distance transport as sources of MPs in remote areas.

We report the abundances and physiochemical characteristics of airborne MPs and non-plastic particles over the Southern Ocean and Antarctica. We used a unified monitoring method along an East Asia-Antarctica expedition cruise spanning from about 30°N to 74°S followed by µFTIR spectroscopy to determine MPs composition. Morphology played an important role in the number of MPs detected: compared with MPs fragments, MPs fibers had increased migration capabilities and can be transported to Antarctica with high efficiency along the transect. In addition, we find the MPs have the similar long-distance atmospheric transport dynamics to the non-plastic particles, independent of particle composition but dependent on morphologies.

## Results and discussion
### Significant latitudinal gradients in airborne microplastics
MPs were detected in all atmospheric samples including those along the cruise path from the mid-Northern Hemisphere (~30°N, close to Changjiang Estuary) to Antarctica (~74°S), as well as those collected in inland Antarctica, ~520 km from the coast (Fig. S1, Table S1). Since all MPs samples were collected in the same manner along a long transect, this makes the entire dataset internally consistent and subsequently the near-global scale distribution of airborne MPs is more robust than comparing MPs collected from different studies. Trends in airborne MPs abundance show that MPs fibers have a constant number of concentrations across a long trans-hemispheric distance but those for MPs fragments decreased dramatically along the cruise path from north to south (Fig. 1). Representative photographs and micro-FTIR spectra of the MPs fragments and fibers are shown in Fig. 2. Similar spatial patterns were also observed for non-plastic particles (i.e., cotton, $SiO_2$, plant seeds, etc.; Fig. 1e, f); that is, non-plastic fibers remained relatively constant and non-plastic fragments show a clearly decreasing trend with increasing latitudes. These spatial trends could be associated with different atmospheric transport efficiencies between the two particles morphologies.

Continental MPs are hypothesized to be the main source for Southern Ocean atmospheric MPs, and we used the linear regression model to fit as explained in Text S1. We used latitude as a proxy for the impacts of anthropogenic emissions, which are expected to decrease from north to south along the study transect. Thus, the linear decrease trend for fragments is not due to the difference in latitude, but rather the distance from populations. The majority of sources (i.e., land from populated urban centers in Southern Asia and Oceania) in the Southern Ocean are located to the north. However, this latitude trend may not be applicable to other regions, such as an equivalent latitude difference across a continent, because inputs from population centers across the latitudes would be continuous.

The number concentrations of MPs along the cruise path ranged from 0.020 to 0.048 n m$^{-3}$, with a mean value of 0.035 n m$^{-3}$ (Table S2). The results are broadly in line with previous sampling efforts in the West Pacific Ocean[15,16]. Fibers were the dominant morphology (~80% along the cruise path, Fig. S2a), ranging from 0.015 to 0.044 n m$^{-3}$ and with a mean of 0.029 n m$^{-3}$. In contrast, the average number concentrations of MPs fragments were ~2–5 times lower than MPs fibers along the cruise path with every 30° interval in the southerly direction, ranging from 0.003 to 0.012 n m$^{-3}$. The airborne MPs concentrations measured in this study (0.035 ± 0.009 n m$^{-3}$) were not significantly different from previous studies sampled over the open ocean or coasts (2.12 ± 3.84 n m$^{-3}$; $p = 0.243$), but significantly lower than those collected over the land by orders of magnitude (24.77 ± 40.61 n m$^{-3}$; $p = 0.040$) (Table S3, Fig. S3). The mean number concentration of non-

plastic particles along the cruise path was 0.096 n m$^{-3}$, which was around three-fold than MPs concentrations (0.035 n m$^{-3}$; Fig. 1f, Table S2). Fibers also dominated (~80%) the non-plastic particles in terms of number concentrations along the cruise path (Fig. S2b). Besides, seasonal variation was not observed in our study, and no significant difference in MPs and non-MPs concentrations between sampling locations A16–A21 (austral autumn; Table S1) and A13–A15, 22 (austral spring; Table S1) within the latitudes of 60°S–65°S was observed ($p > 0.05$; Fig. S4).

We show that MPs are present in the air over inland Antarctica, with mean MPs concentrations of 0.0055 n m$^{-3}$. This is approximately one order of magnitude lower than that in the MBL (0.035 n m$^{-3}$; Table S2). The inland sampling site, Taishan camp, is a summer-only camp, and is very remote with minimal local sources of MPs. Similarly low concentrations of non-plastic particles (0.015 n m$^{-3}$) were also detected, indicating that only a small number of particles of any composition will eventually be transported in the air over inland Antarctica. While an interesting and important finding, considering only two samples were collected at this location, they were excluded from further discussions of MPs in the MBL.

### Hemispheric scale transport of airborne microplastics
The long-range atmospheric transport of mineral dust particles hundreds of micrometers in size has been previously studied[9], and can make an analogous comparison for similarly sized MPs. The transport distance of particles is intricately linked to their aero sedimentation velocity and vertical upward wind. For compact MPs fragments with effective radius $r$, of ~60–80 µm and a density of ~1.4 g cm$^{-3}$ (e.g., PVC), its aero sedimentation speed $U$, is estimated to be between 1.3–1.5 m s$^{-1}$. That is, for an updraft wind with a vertical component greater than ~1.5 m s$^{-1}$, fragments of similar or smaller masses will become airborne and carried by the wind. Similarly, the lower $\rho_p$ corresponds to lower $U$; e.g., polyethylene (0.91–0.94 g cm$^{-3}$) and polystyrene (0.96–1.05 g cm$^{-3}$), their estimated sedimentation speed is ~1.0–1.2 m s$^{-1}$. As reported, the vertical upward wind (1–1.5 m s$^{-1}$) can occur fairly frequently in convective updrafts[17]. In this case, the particles near the surface would be uplifted and released into the atmosphere. In this way, the sedimentation of the particles is outrun by the updraft. For particles with elongated morphologies, such as MPs fibers, the surface area increases in comparison to fragments of comparable mass. Comparing a fragment of 200 µm ($2r$) and a fiber of 1.0 mm in length ($L$) of similar masses, the fiber would sediment at ~45% the speed of the fragment ($L/2r$ ratio of ~5.0; Fig. 3) and thus become airborne easier. The longer the fiber is, the lower the wind speed needed to carry the particle. These differences in atmospheric transport likely account for the difference in latitudinal patterns between the two MPs morphologies studied here (Fig. 1c, f).

In addition, air mass backward trajectories can provide an estimate of the path that air masses followed before reaching the sampling site, which is useful for exploring source regions. We calculate backward trajectories with the HYbrid Single-Particle Lagrangian Integrated Trajectory (HYSPLIT) model. The air masses sampled at low latitudes originated from the continents (e.g., Southern Asia and Oceania; Fig. S5), with statistically higher number concentrations of MPs fragments than samples collected elsewhere ($p < 0.001$). That is, the air masses passing over the continents may bring MPs into the MBL, and the concentrations subsequently decrease as the distance from the coast increases. In contrast, the sampling sites in the high southern latitudes are less susceptible to the continental sources (Fig. S5). For example, the number concentrations of sealant tar, a complex polymer containing toxic aromatic hydrocarbons, aromatic acids and nitrogen-containing compounds, drastically decreased along the cruise path from ~0.004 n m$^{-3}$ in low latitudes to ~0.0008 n m$^{-3}$ south of 60°S (Fig. S6). Similar phenomena of constant concentrations of non-plastic fibers and decreasing burdens of non-plastic fragments were also

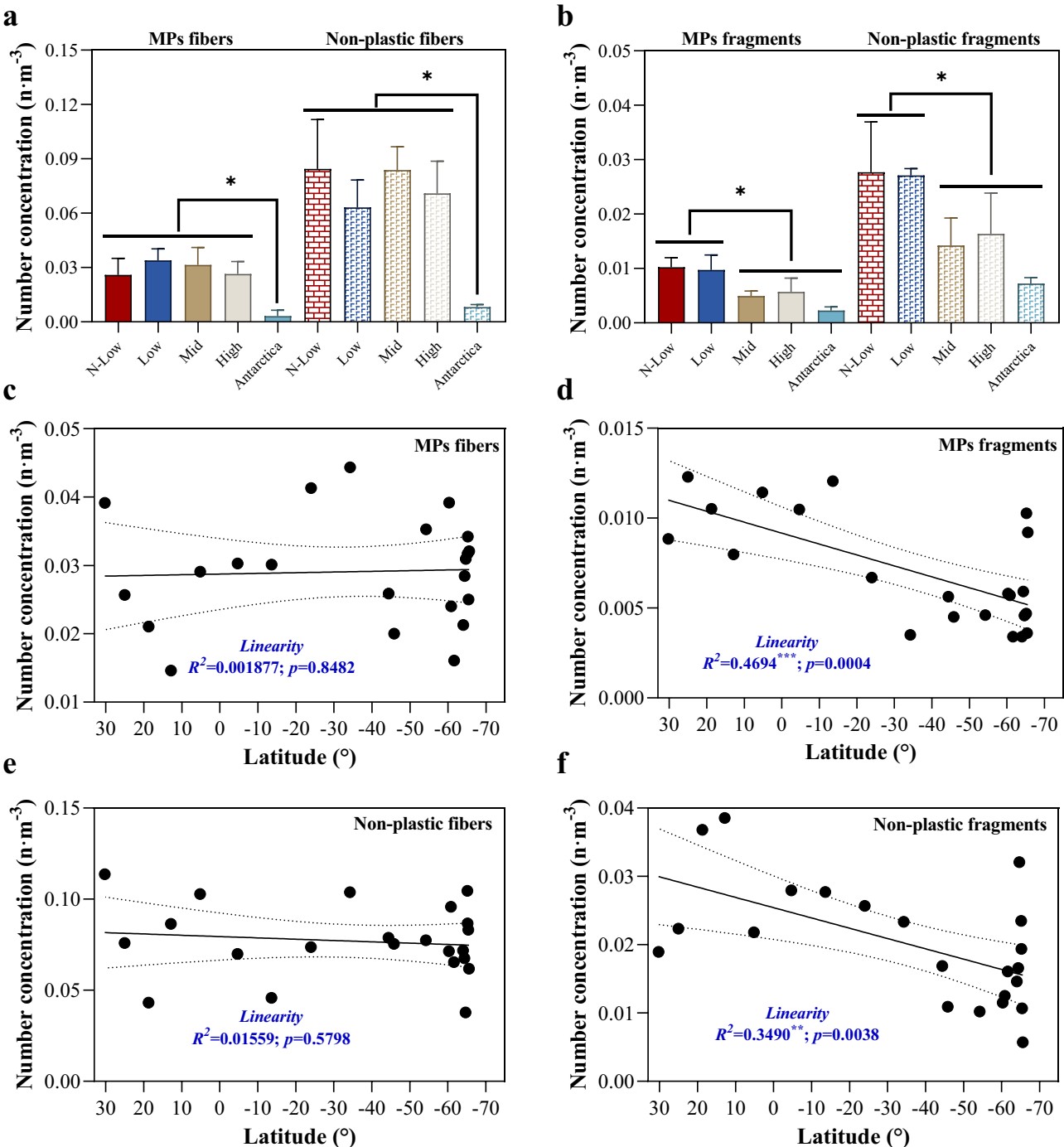

**Fig. 1 | Number concentrations of microplastics (MPs) along the cruise path from the mid-Northern Hemisphere to Antarctica. a** Fibers and **b** Fragments concentrations at low, mid, and high latitudes along the cruise path and over inland Antarctica; **c** MPs fibers, **d** MPs fragments, **e** non-plastic fibers, and **f** non-plastic fragments variations in concentration along the cruise path. *t*-test analysis between the inland Antarctic sites and other latitude regions were performed in panels (**a**) and (**b**), and a stepwise linear regression was used to determine the relationship between particles concentrations and latitudes in panels **c**-**f**. Symbol of * indicates $p < 0.05$. The groups under black lines showed the same statistical analysis results with the comparing groups. Area enclosed by dotted lines (panels **c**–**f**) represent 95% confidence bands. N-low, low, mid, and high latitudes represent the regions of -0–30ºN, 0–30ºS, 30–60ºS, and 60–70ºS along the cruise path. Note that the inland Antarctic sites were excluded in the statistics of marine boundary layer (panels **c**–**f**). Error bars in panels (**a**) and (**b**) represent one standard deviation. The source data underlying (**a**–**f**) are provided in a Source Data file.

observed along the cruise path. Thus, it is suggested that the transport dynamics of MPs are generally similar to those of non-plastic particles in the atmosphere. Given similar densities among particles of different compositions collected along the cruise, we assume that particle morphology largely determines the transport efficiency independent of particle composition. Our results cannot discount the possibility that MnPs are emitted from the ocean as waves break[11,18]. While it is

plausible to consider that fibers can be emitted from the ocean surface and be released to the atmosphere, in a small scale-laboratory study investigating a restricted range of polymer types and fiber lengths (diameters ranging from 25 to 40 μm with an average length of 100 μm), fibers were not emitted in significant quantities compared to spheres and fragments[18]. This may be because fibers have increased surface tension compared to other MPs morphologies. Hence, we

assumed that long-range atmospheric transport was responsible for most of the fibers collected in atmospheric samples in this study.

### Physiochemical composition of airborne microplastics

Of all the MPs fibers collected, a total of 19 polymer chemistries were detected, comprising rayon (52%), polyester (28%), polyethylene terephthalate (PET, 4%), and polyvinyl propionate (PVP, 4%; Fig. 4a, Fig. S7). The percentage of rayon (~50%) was relatively constant across all latitudes. Besides the dominant presence of the four major polymers noted above, nylon, polyethylene (PE), polypropylene (PP), and

poly (methyl methacrylate) (PMMA) were also present in the MBL. The most frequently detected polymers (rayon and polyester) are widely used textiles accounting for >60% global production. Previous research indicates that terrestrial-based MPs fibers are major sources into the MBL air, with emissions occurring during garment wearing and drying[19–21] and clothes abrasion during daily wear[22,23]. Of note, MPs fibers accounted for 11–37% of all the fibers measured in our samples, which means the abundance of non-plastic fibers (especially cotton, $0.042 \pm 0.011$ n m$^{-3}$) was higher than synthetic polymer fibers (Fig. S8, Table S2).

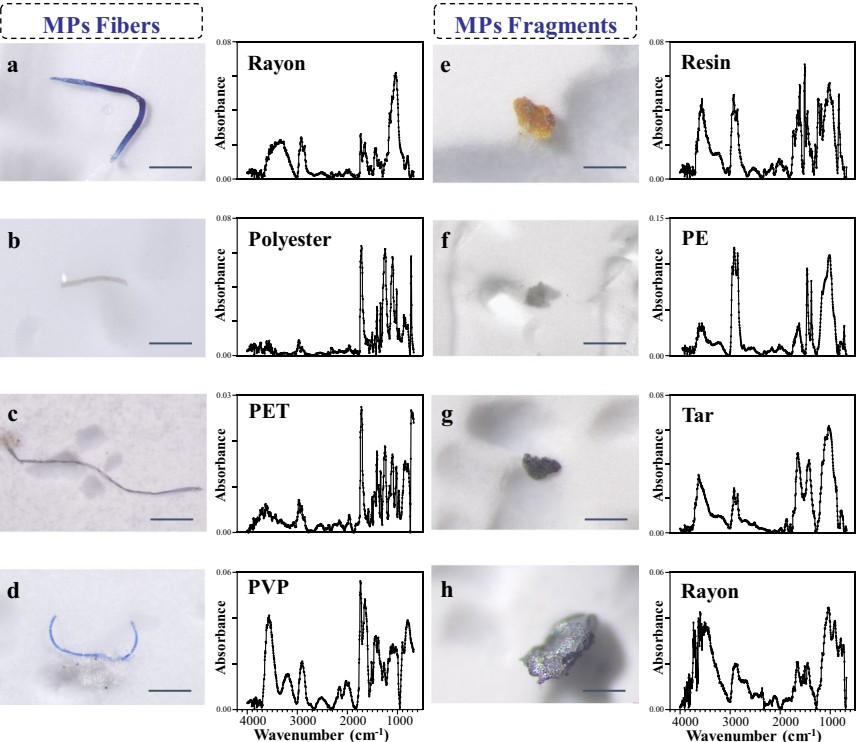

**Fig. 2 | Representative examples of microplastics (MPs) fibers and fragments found in atmospheric samples along the cruise path.** Stereomicroscope images are shown in the 1st and 3rd columns and the corresponding micro-FTIR spectra are provided in the 2nd and 4th columns. MPs fibers (panels **a**–**d**); MPs fragments (panels **e**–**h**). PET: polyethylene terephthalate, PVP polyvinyl propionate, Resin epoxy resin ester, PE polyethylene, Tar sealant tar. Scale bar: 200 μm. The source data underlying (**a**–**h**) are provided in a Source Data file.

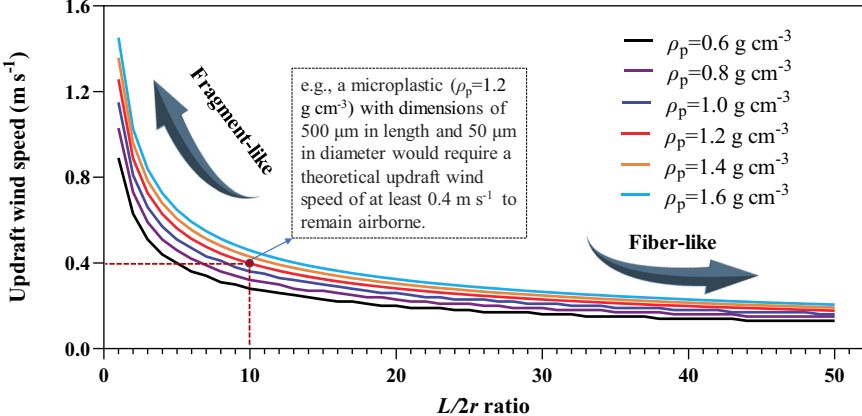

**Fig. 3 | Changes in the vertical component of the updraft wind speed which can carry microplastics with different lengths and densities.** $L$ is the length of a particle and $2r$ is its diameter or width; and $L$ is defined to be always larger than $2r$. The smaller of the $L/2r$ ratio is, the shape of the particle is closer to a fragment; whereas the larger of the $L/2r$ ratio is, the shape of the particle is closer to a fiber. $\rho_p$ indicates the density of a plastic particle. The source data underlying Fig. 3 are provided in a Source Data file.

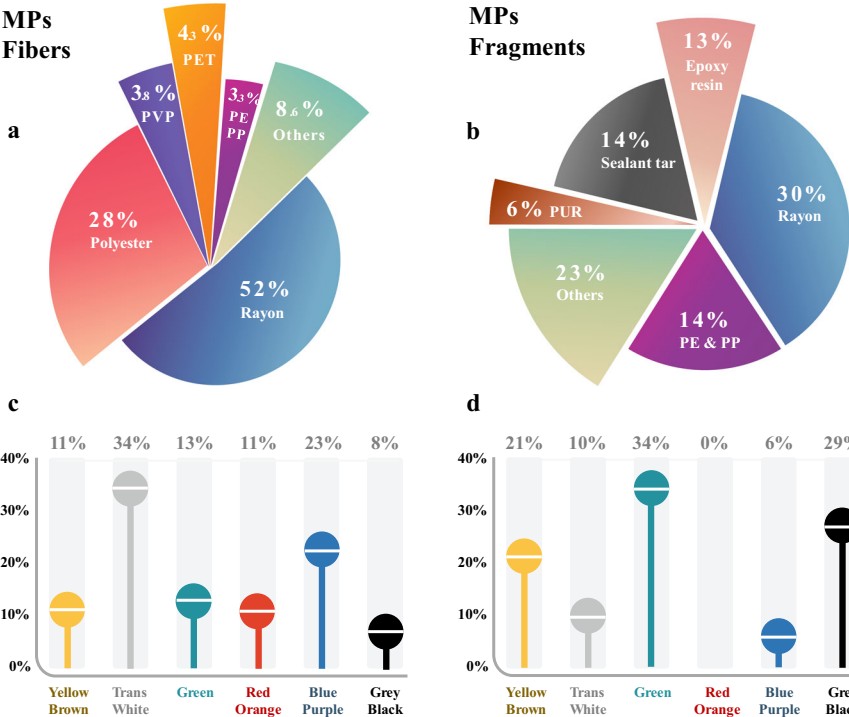

**Fig. 4 | Polymer composition and color of microplastics (MPs) detected in the marine boundary layer along the cruise path. a** MPs fibers chemical compositions; **b** MPs fragments chemical compositions; **c** MPs fibers color; **d** MPs fragments color. The white lines in the center of the colored circles in panels (**c**) and (**d**) represent the mean values of the percentages of the respective colors. PET polyethylene terephthalate, PVP polyvinyl propionate, PP polypropylene, PE polyethylene, PUR polyurethane. The source data underlying (**a**–**d**) are provided in a Source Data file.

For MPs fragments, a diverse set of 16 polymer chemistries were identified across all samples (Fig. 4b, Fig. S9). Compared to MPs fibers, the patterns of MPs fragment compositions were more variable along latitude gradients. Generally, rayon and epoxy resin, widely applied as coatings and adhesives[24], were commonly detected in all samples. Notably, sealant tar accounts for a small amount in plastic production in terms of tonnages produced[25,26], yet its abundance in the airborne MPs fragments reached up to 14%; that is, the proportion of sealant tar in the MBL is higher than nominal polymer production values. Sealant tar plastics usually contain complex toxic compounds, and thus its application (e.g., pavement sealant tar) has been banned in some regions (e.g., Texas, USA)[27]. However, they are still widely used in dams sealing worldwide[28,29], and sometimes they may also be present in ships' coating[30]. To verify this, several dam sealant particles collected along the East China Sea coast (31.0°N, 121.9°E) were identified and their FTIR spectra matched with sealant tars found in the MBL well (Fig. S10), indicating that air mass trajectories from winds over these regions could be an important source of sealant particles. The color of MPs fibers was similar along different latitudes, where white/transparent MPs were the most dominant (34%). Conversely, MPs fragments exhibited different colors along the cruise route (Fig. 4c, d, Fig. S11).

Neither the length or diameter distributions of MPs fibers or fragments recovered from the MBL showed clear spatial trends (Fig. 5a, b). The lengths of the MPs fibers were 1004.8 ± 823.5 μm and their diameters were 49.8 ± 39.5 μm. Notably, the size dimensions and densities of non-plastic fibers and MPs fibers were quite similar (Fig, S12, Table S4), where non-plastic fibers lengths and diameters were 825.2 ± 723.3 μm and 51.4 ± 39.1 μm, respectively. For MPs fragments, particle lengths were 166.2 ± 88.6 μm, with widths of 99.3 ± 54.4 μm (Fig. 5d, e). Similar physical properties, that is length, width, and density, of non-plastic particles were also observed (Fig, S12), indicating that transport is independent of particle composition, as the average particles densities of major components (rayon, polyester, and resin) were within the range of 1.2–1.5 g cm⁻³ for particles in this size range.

The friction drag force is positively correlated with particles' surface area[31], and thus it plays an important role in the transport dynamics of aerosols. The surface area of MPs fibers (median = 0.17 mm²) was approximately twice that of MPs fragments (median = 0.077 mm², Fig. 5c, f), which indicates that the transport efficiency of MPs fibers would be higher than that of MPs fragments, with the assumption of similar densities (Table S4) and masses. The masses of these two MPs morphologies in the samples we collected were close, with 2.92 μg and 2.34 μg for MPs fibers and MPs fragments on average along the cruise path, respectively (Fig. S13, Table S5); whereas due to their different surface areas can lead to different drag forces (Fig. 5c, f), they exhibit very different transport efficiency.

## Physicochemical weathering of microplastics during atmospheric transport

During the atmospheric transport of MPs, physiochemical weathering of MP particles can occur[32], which begins with ultraviolet irradiation and oxygen interactions, causing changes to polymer functional groups, mechanical properties, and surface roughness[32,33]. One indication of aging is changes of the polymer carbonyl index, which is one of the most common assessments to measure the chemical oxidation of polymers. We observed a higher carbonyl index for MPs collected from the MBL at higher latitudes compared to mid-low latitudes (Fig, S14), and the change of the C = O position (at 1735 cm⁻¹) was clearly shown (Fig. S15), suggesting increased weathering during long range transport. We chose rayon for a cross-latitude carbonyl index alteration analysis, because rayon (man-made cellulose) which has a large detection rate, does not contain C = O itself, and has been verified to be photochemically degraded by near UV and visible radiation and form oxidized group (carbonyls and carboxyls)[34–36].

In addition to the formation of oxygen-containing groups, physicochemical weathering can also generate peeling and drag deformation or cracks and holes on MPs (Fig. S16), thereby increasing particles' specific surface area[37]. Surface microcracks can increase

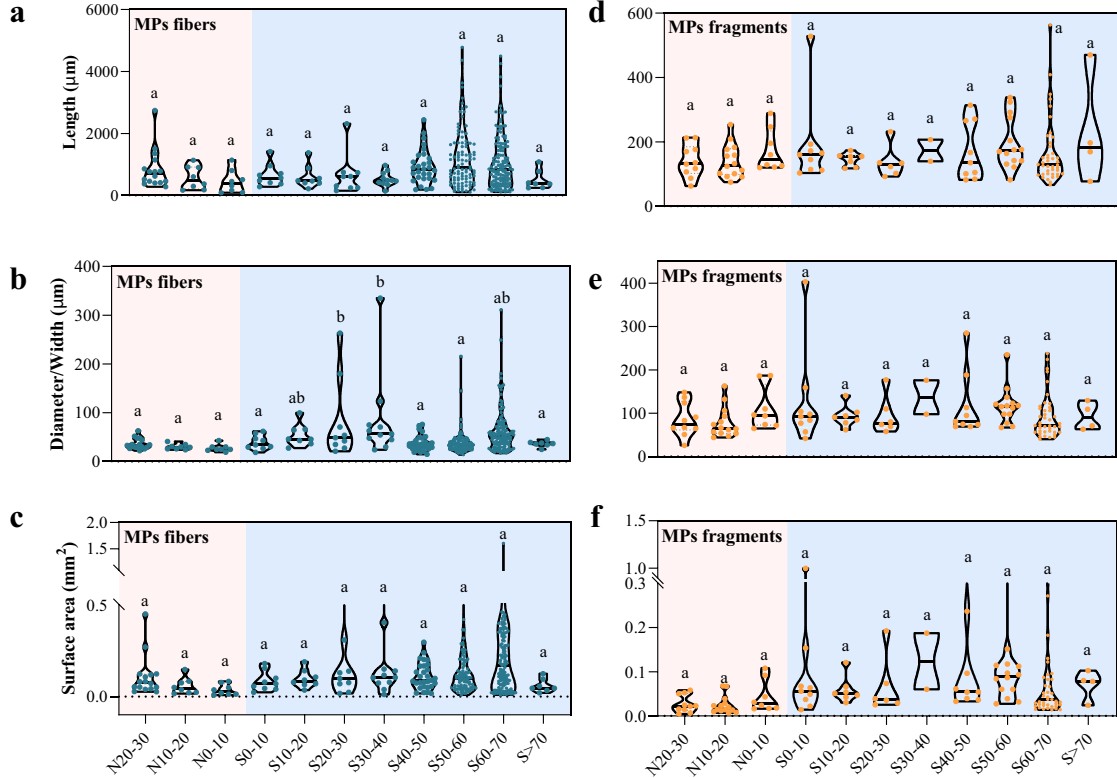

**Fig. 5 | Dimensional analysis of microplastics (MPs) fibers and fragments.** **a** Fiber length; **b** fiber diameter; **c** fiber surface area; **d** fragment length; **e** fragment width; **f** fragment surface area. N and S indicate the latitude of the north and south, respectively. S > 70 indicates inland Antarctica. Red shaded areas indicate the Northern Hemisphere, and blue shaded areas indicate the Southern Hemisphere. The surface areas were calculated based on the particle dimensions and are not referring to the specific surface areas. Letters above violin plots correspond to statistical comparison by one-way analysis of variance (ANOVA): groups with at least one same letter indicates no statistical differences, while groups marked with different letters indicate significant differences ($p < 0.05$). Black lines cross the plots are median values. The source data underlying (**a**–**f**) are provided in a Source Data file.

migration resistance and therefore also reduce transport efficiency. However, weathering can also cause changes in the mechanical properties of MPs, for example, chain scissions triggered by weathering decreased the density of entanglements in the amorphous phase of MPs[38]. Moreover, weathered MnPs in the atmosphere may lead to increased cloud condensation nuclei or ice-nucleating particles in cloud formation processes, especially in areas where there are fewer other aerosols present (i.e., Southern Ocean), and thus could play an important role in cloud formation processes[39]. MnPs weathering can also facilitate the release of additives and associated environmental pollutants once they are inhaled by organisms and diffuse into biological fluids following fugacity ladders, and thus potentially also influencing the biogeochemical behaviors of co-existing pollutants in the atmosphere.

## Implications and outlook

This study applies consistent methodology to sample and measure MPs over a large geographical area, including over the remote Southern Ocean (south of 60°S), and thus the burdens of MPs (>20 µm) in the MBL can be more precisely calculated and compared consistently in the context of one sampling campaign. Average number concentrations of MPs fibers and fragments over the Southern Ocean were $0.029 \pm 0.008 \, \text{n m}^{-3}$ and $0.007 \pm 0.003 \, \text{n m}^{-3}$, respectively, which is an important report of MPs in this region. If it is assumed that MPs are mostly distributed near the ocean surface and decrease with altitude in the troposphere (about 10 km), similar to previous studies[9,40], it is estimated that $(3.7 \pm 1.0) \times 10^{15}$ MPs fibers and $(8.3 \pm 4.1) \times 10^{14}$ MPs fragments are in the atmosphere over the Southern

Ocean (an area of ~$2.2 \times 10^{7} \, \text{km}^2$). With the mass concentrations of 65 ng m$^{-3}$ (MPs fibers) and 7 ng m$^{-3}$ (MPs fragments), the burdens of these two MPs morphologies over this region are estimated to be $8.4 \times 10^{9}$ g and $0.9 \times 10^{9}$ g, respectively. The total burden of MPs over the Southern Ocean, about 10–11 thousand metric tons, generally accounted for a small fraction of the previous estimates on global marine atmospheric burden of MPs (0.013–25 Mt)[7]. Consequently, the observations here provide a benchmark for current global MPs pollution in these regions, which is essential for assessing the effects of future measures on plastic production and treatment.

While the overall climate impacts of airborne MPs have yet to be quantified, the estimated mass concentrations of MPs over the Southern Ocean (Table S5) are orders of magnitudes lower than the concentration of sea spray aerosols (e.g., Cl⁻ and Na⁺ of ~9000 ng m$^{-3}$), and thus the MPs may play a minor role in the marine cloud nuclei[39,41]. Though the number concentrations of MPs are very small and may not have discernible influence on radiative forcing, we found that most of the MPs are not transparent white, and thus the direct radiative effects of these colorful MPs still deserve further investigation. In addition, MPs serve as chemical carriers that can release plastic additives and absorb air pollutants, transporting co-existing chemicals to remote regions, which implicates uncertain ecological outcomes.

## Methods
### Sample collection
Airborne MPs were collected along a Chinese Antarctic expedition cruise path during the 2019-2020 season (Fig. S1a). Atmospheric samples were collected with a high-volume air sampler (HVAS, TISCH

Environmental, USA). The sampler was situated on the top deck of the RV/Xuelong ~25 m above the sea surface. Atmospheric particles were collected onto Whatman quartz fiber filters (QM-A, 20.3 cm × 25.4 cm, with the pore size of 2.5 μm). At each site, the membranes were pre-baked at ~500 °C for >6 h to ensure the filters were free of polymer particles. For mounting the filters, the buckles of the upper placing plate were unscrewed after opening the upper cover of the HVAS, and then a filter membrane was laid flush on the lower plate. Then the buckles were re-tightened, and the upper cover of the HVAS was closed. The entire process was generally completed within about 30 s, reducing the chance of airborne contamination. Note that the filter has one rough surface and one smooth surface, and atmospheric particles were always collected on the rough surface. Typically, the pump of the high-volume air sampler (TISCH Environmental, USA) was running at a constant flow rate, 1.2 m³ min⁻¹, and for a sampling duration of 48 h which would consequently lead to sampling a volume of 3456 m³. The flow rate of the pump was calibrated when manufactured in the factory. Atmospheric particles along the 2 days' cruise path, covering ~2–4 degrees of latitude, were collected on one filter which was considered one aggregate sample. A wind direction sensor was employed to control the HVAS to avoid potential contamination from the vessel, so consequently only air masses from a sector ~120° left and right of the central line of the vessels' path was sampled. After sampling, the upper cover of the HVAS was opened and buckles of the upper placing plate were unscrewed, then the filter was removed by pre-cleaned stainless tweezers from the sampler. Individual filters were kept separate, folded, wrapped in aluminum foil, placed in zip-loc bags, and stored in the dark at −20 °C until particle characterization processing began. In total, 24 samples were collected in the marine atmospheric boundary layer from the mid-Northern Hemisphere (~30°N) to East Antarctica (~70°S).

In addition to the marine atmospheric samples, two atmospheric samples were collected from inland Antarctica near the Chinese Taishan camp (73.9°S, 77.0°E), which is ~520 km from the coast (Fig. S1b). Furthermore, four field blank samples (two collected along the cruise path and two collected inland) were prepared from filters mounted in the HVAS with an air pump flow rate set to 0. To avoid the potential contamination from the surrounding air environment, the period of time for the blank filters was set as "30 s". For the field blanks, no MPs were found. The sampling protocols were the same as those described above in terms of sampling duration, filter mounting, collection, transport, observation, and measurements. More details on inland Antarctic atmosphere sampling can be found in our previous work[42].

## Microplastics quantification and characterization

Upon delivery to the laboratory, filters were carefully unpacked in a class 100 clean hood (Sujie Inc., China) to avoid potential ambient air contamination and cut with stainless steel scissors which were rinsed between samples with Milli-Q water. The filter pieces were then transferred to clean glass Petri-dishes. For sample analysis, one fourth of the filter was used for MPs characterization according to a previous airborne particulate study[43], while one half was used for sea salts. Sea salt aerosol concentrations (i.e., Cl⁻ and Na⁺) were analyzed with an Aquion RFIC ion chromatograph (IC, Thermo Scientific, USA)[44]. The remaining one fourth filter was archived and used for method verification during the second-round analysis (Text S2). As only one-fourth of the filter was used for MPs quantification, final number concentrations were derived by multiplying the measured values by four.

The analytical approach we used in this study is in line with approaches which are typically used for collection and characterization of airborne microplastics, despite there being no standardized methods across studies. All particles (>20 μm) on filter membranes were observed using a stereomicroscope (Discovery V8, Carl Zeiss,

MicroImaging GmbH, Göttingen, Germany) and photographed with an AxioCam digital camera coupled to the microscope. All particles were separated according to morphology, with a distinction between fibers and fragments. Each particle's dimensional information was obtained using the ImageJ software (ver 1.8.0, NIH, USA), including length, diameter/width; and surface area values for fibers and fragments were estimated. In addition, a detailed distinction of all particles' colors was made according to the Pantone Color Card (1110 colors) and were later subsequently categorized into eleven categories including transparent, white, gray, black, blue, purple, red, orange, green, yellow, and brown.

The chemical compositions of all particles were analyzed with micro-FTIR using transmission mode (Nicolet iN 10, Thermo Fisher, USA). Each particle's chemical composition was identified with a matching index ≥ 70% to commercial spectra libraries similar to our previous work[45] and combined with expert identification according to polymer characteristic peaks. Non-plastic particles were those identified as non-plastic (e.g., cotton) or materials that could not be recognized by the commercial FTIR spectral library. Considering that the man-made fiber rayon is challenging to distinguish from cotton or other cellulosic materials, we have adopted two methods for its accurate identification. First, the characteristic peak at 1105 cm⁻¹ is very distinct for rayon, and we have embedded this information into the spectra matching library according to our previous research[46]. Second, another characteristic band at 3330 cm⁻¹ of the O−H stretching can also be utilized to distinguish rayon, where its band is rather broad and featureless, whereas it oppositely exhibits a distinct maximum at 3330 cm⁻¹ for natural fibers[47].

All individual fragments collected across all field samples were characterized. For fibers, we found that 14 samples in the more northerly sample locations had too many fibers on one fourth of the filters to easily measure (number of fibers exceeded one hundred). To facilitate the identification and quantification, all fibers were observed and collected under the microscope and total numbers were counted. Then, we randomly selected 30% of the fibers on these 14 filters to identify their composition chemistries. For this method verification, we used the backup samples (another one fourth of the filters) during the 2nd analysis for a full scan to prove the robustness of the sub-sampling approach (i.e., only using 30% of the fibers in the 1st analysis) by comparing these values with the entire analysis (100% of the fibers analyzed in the 2nd analysis) (see Text S2). Therefore, we identified all fibers from the first 12 sampling sites and 30% (the 1st analysis) or 100% (the 2nd analysis) of the fibers from the subsequent 14 sampling sites, and adjusted the final fiber number and chemical identification reported accordingly. Daily laboratory blanks were performed during the chemical identification experiments. The daily laboratory blanks and procedural blanks were analyzed in the same manner across the entire particle identification processes. The purpose of these blanks was to quantify the possible dust (including MPs) contamination caused during the entire workflow. For all the daily and procedural blanks, no MPs or non-plastic particles were found.

## Conversion of microplastics and non-plastic particles size to mass

To estimate the MPs and non-plastic particles airborne flux, we also need to know the mass concentration (MC) in addition to the number concentration (NC). We used the cylinder model[48] and the column model[49] to estimate the surface area and volume (mass) values of fibers and fragments (Text S3). In addition, an approximation by assuming the length-to-width ratio equates to the width-to-height ratio ($L/W = W/H$) was also applied for fragments[48,50]. However, it is important to note that this simplified approach to suggest surface area and volume (mass) likely represents a low estimate, as it does not take into account additional surface roughness and cracks which are likely to be present on environmental MPs, subsequently increasing the actual

surface area. To calculate the MC of individual airborne fibers (Eq. 1), we approximated fibers to be cylinders and considered a void fraction (40%)[48], where $n$ is in the total number of fibers in the sample $i$; $v_i$ is the sampled air volume; $f$ is the fiber void fraction (40%), because airborne fibers become looser than their original states (Fig. S16), and the value of $f = 0.4$ was used here. $R$ and $L$ were the fiber diameter and length as calculated by the ImageJ software according to the top-view projection images, respectively. $\rho$ is the average density of primary fragments collected in this survey (Table S4).

$$MC_{fiber} = \sum_{k=1}^{n} (1 - f) \cdot (R_k^2 \cdot L_k) \pi \rho / v_i \quad (1)$$

$$MC_{fragment} = \sum_{k=1}^{n} S_k \cdot (W_k^2 / L_k) \rho / v_i \quad (2)$$

To calculate the mass of airborne fragments, we assumed they were smooth fragments and used the Eq. 2. $S$ was calculated as the width times length of each fragment according to top-view microscopy images. $H$ is the height of a fragment, which was assumed as $W^2/L$, according to the approximation by Koelmans et al.[49].

## Particle weathering index calculations

To quantify the degree of photochemical weathering during the transport, a common metric carbonyl index (CI) which measures the growth in the carbonyl (C = O) absorption of MPs[51] were calculated using Eq. 3, which represents the absorbance ratios for the C = O detection wavelength band, and C-H$_3$ detection wavelength band[52].

$$CI = A_1 / A_2 \quad (3)$$

where, $A_1$ is the absorbance of the carbonyl (C = O) peak for MPs, and $A_2$ is the absorbance of a reference peak (CH$_2$) for these MPs samples[53]. Detailed area bands were depicted in Text S4 and Fig. S15.

## Estimation of particle updraft speeds

When the speed of an updraft, in particular its vertical component, exceeds the sedimentation speed of an object, the object becomes and remains airborne. The wind-carrying speed for a compact fragment, simplified as a sphere of radius $r$, can be estimated through the balance between the aerodynamic form drag it experiences and its body weight (Eq. 4):

$$\left(\frac{1}{2}\right) C_D \rho_A A U^2 = \left(\frac{4}{3}\right) \pi r^3 \rho_p g \quad (4)$$

where $C_D$ is the drag coefficient of order 1; $A$ is the wind-facing projection area; $U$ is the air-relative speed; $g$ is the acceleration due to gravity; and $\rho_A$ and $\rho_p$ are the air density and plastic density, respectively. The left side of the equation is the aerodynamic form drag. The weight of the fragment $W_p$, is expressed on the right. Letting $A = \pi r^2$, the minimum drafting windspeed for a compact body can then be estimated. For instance, when $r \sim 65\,\mu m$ and $\rho_p \sim 1.4\,g\,cm^{-3}$, wind speed $U$ is found to be ~1.4 m s$^{-1}$. However, if this fragment is stretched and becomes a fiber, its weight is fixed but takes on an elongated geometry, the right side of Eq. 4 is unchanged but area $A$, on the left side is increased. Consequently, a slower wind speed, $U$, is needed to balance the two sides. This implies that fiber-like particles can be suspended in the air and be carried more easily than the compact, spherical ones.

Given the size and weight observed, speed $U$ can be obtained through Eq. 4. The corresponding Reynolds number (Eq. 5), which measures the relative magnitude between the inertia force (form drag) and viscose force (skin drag), yields Re = 4 ~ 17. Since it is well above 1,

the use of the above form-drag formulation for windspeed estimation is justified.

$$Re = 2rU / \upsilon_{air} \quad (5)$$

where $\upsilon_{air}$ is the kinematic viscosity of air ($1.48 \times 10^{-5}$ m$^2$ s$^{-1}$ or 14.8 cSt at 15 °C and 1 bar), and the parameters $r$ and $U$ are the same as those in Eq. 4. The fragment weights $W_p$ in this study are observed to be ~3.0 μg, with the average $r$ and $\rho_p$ of 0.13 mm and 1.4 g cm$^{-3}$, respectively. From Eq. 4, the wind speed $U$ is estimated to be 1.3–1.5 m s$^{-1}$. For a fiber, its surface area $S_{fiber}$ is greater than area $S_{frag}$ for a compact fragment by a factor $\Gamma$. This factor $\Gamma$ is related to its length change by:

$$\Gamma = S_{fiber} / S_{frag} \sim (L/2r)^{1/2} \quad (6)$$

where $L$ is the length of the fiber and $2r$ is the diameter of the compact fragment. The square root dependence on $L$ can be readily seen when one stretches a cubic solid of fixed volume into a long bar of length $L$ with two shrinking square cross-sections. Comparing a compact plastic fragment and a plastic fiber of similar weight, the wind-carrying speed will be smaller for the fiber since its area is greater. In another word, fibers are easier to be airborne than compact fragments.

## Back trajectories analysis

Air mass backward trajectories can provide an estimate of the path that air mass followed before reaching the sampling site, which is useful for exploring the possibility to source track atmospheric particles. Here, we calculated the 7-day backward trajectories arriving at the sampling location with the aid of the HYSPLIT, developed at the Air Resources Laboratory of NOAA[54]. All back-trajectories were calculated using the National Center for Environmental Prediction-National Center for Atmospheric Research (NCEP–NCAR) reanalysis data (1°×1°; 17 vertical levels). The vertical velocity model was used for the calculation of the vertical motion of air mass. The height of the end-point of trajectories (i.e., the sampling location) is set to 500 m above mean sea level. 7-day back trajectories were run for the time of sampling (Table S1). The length of 7 days was chosen because previously published reports suggested that the residence time for MPs in the atmosphere may vary between 1 and 156 h[55], and the backward trajectories were usually run for 6–7 days to show the full range of possible sources[56]. It is noted that the calculation of backward trajectories is under the assumption of no vertical mixing at night and complete mixing over the planetary boundary layer in the daytime.

## Data analysis and statistics

All maps were generated using ArcGIS10.2 tools (ESRI Co, Redlands, USA) and all graphs were either generated using GraphPad Prism 9 (GraphPad Software Inc., USA) or Origin 9.0 (OriginLab Corp. USA). Statistical analysis was carried out using the software, SPSS 22 (SPSS Inc. USA). Data were reported as mean ± SD. Data normality was tested by the Shapiro-Wilk's test. If the distribution was normal, one-way ANOVA and $t$-Test analyses were conducted to determine differences among groups (>3) or between two specific groups. If not, nonparametric analyses followed by the Krustkal–Wallis test were used. Linear regression analysis was conducted using SPSS 22.

# Data availability

All data supporting the findings of this study are available in the article, supplementary information, and source data. Source data are provided with this paper.

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

## Acknowledgements

This work was supported by the National Key Research and Development Program of China (2022YFC3105900 to Q.C.), the National Natural Science Foundation of China (NSFC) (42276243 and 41922046 to G.S., 42077371 to Q.C., 92252204 to J.Z.), the Program of Shanghai Academic/Technology Research Leader (20XD1421600 to G.S.). D.M.M. was funded through the Swiss National Science Foundation, grant number PCEFP_186856. L.E.R. and E.C.L.R. were supported by the Royal Society of New Zealand Marsden Fund (MFP-UOC1903 and VUW2118). L.E.R. appreciates support by the Rutherford Discovery Fellowships from New Zealand Government funding, administered by the Royal Society Te Apārangi. The authors are grateful to CHINARE members for their support and assistance in atmosphere sampling.

## Author contributions

Q.C. contributed to the experimental design, micro-FTIR analysis of samples, writing original draft, review and editing; G.S. contributed to the experimental design, cruise routes design, backward trajectories analysis, writing original draft, review and editing, supervision; L.E.R. contributed to the writing, review and editing; J.Z. contributed to the updraft wind speed calculation, writing, review and editing; C.Z. contributed to the micro-FTIR analysis of samples; D.W. contributed to the atmosphere samples collection; G.W. contributed to the membrane pretreatment, samples separation; E.C.L.R. contributed to the writing, review and editing; D.M.M. contributed to the writing, review and editing. All authors read and approved the final manuscript.

## Competing interests

The authors declare no competing interests.
