## [Peer Review File · Nature Communications]

Long-range Atmospheric Transport of Microplastics across the Southern HemisphereReviewer #1 (Remarks to the Author):

I read the manuscript "Long-range Atmospheric Transport of Microplastics across the Southern Hemisphere" dealing with the atmospheric transport of microplastics to remote regions. The authors present a hemispheric-scale analysis of airborne microplastics taken during a cruise path. It is true that the transport efficiency of microplastics in the atmosphere is still poorly known, not only in the Southern Hemisphere, but globally, and the data presented are interesting. However my feeling is that they are more routine than a real breakthrough: the results showed that fibers were more efficiently transported, which is a known fact. Besides, the discussion provided by the authors and the interpretation of data present some shortcomings. In include details below.

I think there is a factual error in line 56 concerning the cite of D. Allen. The size range is not 0.2-5000 μm , but 20 μm -5mm: "However, this marine sampling comprises particles collected predominantly in the range of 20 μm -5mm".

0.2 μm is already in the nanoplastic range and mentioning this size range is risky. Currently, there are no techniques for detecting nanoplastics in environmental samples except in very specific cases. It is true that some authors reported high concentrations of nanoplastics, even much higher than the concentration of most pollutants in wastewater, which makes them unbelievable. I know this is the way to gain headlines, but extraordinary claims require extraordinary evidences.

The authors claim (several times) that they used a unified and standardized monitoring method, but this was because they were the results of a single expedition. It would have been strange that they didn't unify methods with themselves. Comparing data from different studies is not a weakness (line 82), but a strength because of result cross-checking.

Lines 151-152: Reference 14 does not refer to fibres.

Concerning chemical characterization, regenerated cellulose is very difficult to distinguish from cotton or other cellulosic materials using spectroscopic methods. As rayon dominates the composition of the samples taken in this work this is not a minor issue as natural fibres have been shown to dominate airborne fibres in other studies. The authors should discuss the details of the spectra in the region 3000-3800 cm^{-1} that support the attribution to rayon of a high number of fibres.

The discussion in the paragraph starting in line 198 (and Ref. 26) refers to equivalent diameters, which is a concept absent from the current manuscript. Beside, ImageJ cannot obtain surface area. The authors probably mean projected area.

What is the pore size of glass fibre filters and how the authors measured the flow rate? Didn't the flow rate change during sampling? This is the reason why sampled volumes are generally much lower than those reported in this work. Sampling details are not sufficiently detailed in the manuscript.

Carbonyl index is sometimes used as a measurement of photochemical ageing, but it is cumbersome to assume that it will systematically increase during long-range transport, which can last hours or a few days, depending on the atmospheric circulation. Besides Fig. 5 is non-conclusive and does not support authors' opinion. Additionally, I don't see any peak in the 1735 cm^{-1} region in the rayon spectrum of Fig. 2, where C=O stretching band should be visible.

In sum, although the authors presented interesting results, neither their novelty nor the depth of discussion is sufficient to grant publication in a top journal like NC.

Reviewer #2 (Remarks to the Author):

Dear authors, you present noteworthy results of the presence of MPs in the atmosphere across the Southern hemisphere. your work is of significance to the field of atmospheric MP research. Some information in the methods and the

subsequent use of the acquired data does not always support the conclusions and claims, with additional input and revision needed?

Some flaws in the data analysis, interpretation and conclusions are pointed out below in more detail, requiring revision.

Line 104: Please also see comment on method chapter, but the described use of only 30% of 1/4th of the filter results in 1.4 and less MPs per sample. The reanalysis of 100% of the 1/4 of the filter is strongly advised for all samples inside the MBL.

Line 118: The use of a density of 1.4 g/ cm³ covers only some polymers, with the most prevalent ones in terms of global production and use are characterized by much lower densities below or close to 1 (PE, PP, PS). State alternative sedimentation rates for these polymers as well.

Line 154 ff:

Rayon is a semi-synthetic polymer, consisting of cellulose. The discussion of the ability of the FTIR to distinguish between natural cellulose fibres and rayon is missing. As a consequence, the inclusion of data on rayon should be limited to general descriptions rather than detailed data evaluation and discussion.

Line 209 ff

Since the weathering evaluation was only done with rayon with its challenges to be correctly distinguished from natural cellulose, and rayon being the almost only polymer with a considerable number of CH₃- and C=O groups potentially formed, this part is highly speculative and not supported by the available data. Please remove.

Line 439:

Please add also metric dimensions, since all other sizes are given in metric units.

Line 456: State the period of time the fieldblank was placed in the sampler.

Line 461 ff:

Please extend on the information on how the filter was placed, removed from the sampler and stored prior analyses

Line 493:

Please specify if the analysed MPs represented 30% of the 1/4 of the filter? Can you show that this is representative for the whole filter also from filters from inside the MBL with a low MP count?

Line 502:

What about the fieldblanks? these are the important blanks that should be used in the QA/QC of the data evaluation. Please add information on the fieldblank MP content and how the data was used to blank correct the field data.

Line 516.

A reference for this method is missing. Was this method tested with spiked samples with known weight? Which mean density was used here?

Line 523:

A reference for this method is missing. How was that method verified?. State that this a method only applicable for rayon and other polymers forming C=O from CH₃- groups

Reviewer #3 (Remarks to the Author):

The paper analyses the microplastic concentrations in the atmosphere over the marine surface collected during a cruise in the Southern Ocean. The study gives important results on the field, providing with a new relevant dataset of observations in a region that was otherwise not well explored yet. While, for the relevance of the results and content of the paper I would encourage a final publication of this work on Nature Communications, I think some major revisions on the way

the data, the analysis, and some of the conclusion are presented, are necessary.

I summarize my comments and suggestions here below:

The main weakness to the data presentation is the lack of any temporal reference, as there is no info on when the observations have been collected. It is only mentioned on the method paragraph that the campaign took place in the "2019-2020 season". This is very vague, and it doesn't allow for any consideration on the atmospheric processes that may have influenced the MP transport. This information is also missing from the data files provided in the attachment. As this will be very important to understand the atmospheric conditions leading to the MP transport, and to reproduce the results of the study, it is essential that all the data also reports the representative date of collection for each sample. I do encourage putting this not only in the uploaded data, but also in the tables S1 and S1 and the figures (for example in the S4).

Also, this raises important questions on the possible interpretation of the collected data, for example: are the different samples collected at different latitudes related to different seasons? This need to be considered to understand the transport conditions and the representativeness of the data.

Lines 84-85: The authors suggest that there is a decrease in MPs fragment concentration from north to south, as if there is a linear relationship between latitude and microplastic presence in the atmosphere (hypothesis also reinforced by the use of the linear regression analysis in figure 1) which do not seem to be justified. What is this hypothesis based on?

Lines 133-136: The back-trajectory analysis, which seems to carry an important message in this work (i.e. that the northern latitudes samples are related to a meaningful influence of land sources with respect to the southern ones), looks not very robust. My main concerns are the following:

- it is not clear how the trajectories release points (i.e. the samples for which the transport analysis is shown) in figure S4 are chosen. Why not putting together the air masses transport analysis for the different latitude sections instead of doing the analysis only on 6 samples?

- Panel B is cut on the northern latitudes and it is not really possible to see if the trajectories are actually reaching the land or not

- The trajectory analysis does not seem to consider the vertical uplift of the air masses. That means that it is not considered if the air masses when traveling over land are reaching the surface or are above the planetary boundary layer. If the last condition is true, then the land sources may actually not have influenced the air masses and the MPs could come from the ocean instead.

Also, in the description of the trajectories methods (lines 579,580) is not clear what is meant by "Back trajectories were run every 12 hours during the 7 days before the sampling date". That sounds incorrect, as there would be no reasons to release trajectories up to 7 days before the sampling date. Is 7 days the length of the trajectories computation back in time, or the length of the release time interval? This needs to be clarified and both quantities need to be specified.

Lines 149-152: This is another important concept that needs to be stated cautiously. The results from the study of Yang et al. 2022 are relative to specific sizes of fibers (~100 um length) and it has been performed in a tank experiment. While this is a starting point to hypothesize that fibers are ejected from water with less probability with respect to spheres, it is still not enough to assume safely that the long-range transport is the responsible for the fibers' presence observed in this study. Since the fibers you observe are quite large in diameter (order of tenths um) and length (order of hundreds um and some even thousands), even if the morphology surely affects their lifetime in the atmosphere with respect to fragments, one can also hypothesize that are also not being suspended since too long. Are the back trajectories originating from the fibers sampling always reaching land in a reasonable time frame? And how is this related to the number of fibers observed? This can be a way to check, in case you want to assume that they are only emitted on land, which would be the time of transport from there and if it is more reasonable to hypothesize that they are actually also emitted from the sea.

Lines 168-186: I think the analysis on the sealant tar is quite interesting, but I do wonder how the

authors can exclude that this is not coming instead from ships coating (for example epoxy resins are widely used for this purpose) and therefore from ocean rather than the coast (or a mixture of the two).

Minor and technical comments:

Lines 61,62: In this sentence the authors seem to put together two concepts (the ocean winds/sea spray mist with the earth electric field forces) that I am not sure are really fitting with each other in the framework of that paragraph. If we want to emphasize the uncertainties related to the presence and transport of atmospheric microplastic in the general sense, at hemispheric level (as the paragraph seems to suggest), it is the identification of the sources, the mechanism of emission, resuspension and the transport and removal processes of MPs that constitute the overall uncertainty. While the electric forces may be indeed part of the general equation, the ocean winds and the sea spray represent only a small fraction of it when talking about the total sources. I guess the authors mean that, among all the uncertainties that are related to the MP transport in the atmosphere, the work is going to provide observational quantification of the presence of MP in the Southern hemisphere marine atmosphere, which will help understand better the role of the ocean as a source of MP in the bigger picture. If that is the case, I would encourage the authors to emphasize more clearly this point.

Figure 1: It is not clear to me how should I read the ** symbols over the connected histograms. Why is in panel A connecting all latitudes with Antarctica and in panel B the Northern latitudes with all the others? What does the purple horizontal bar represent?

Lines 144,145: This sentence is not clear, there may be a typo. Do you mean "Similar phenomena OF constant concentrations...?"

Lines 184-186: Do you mean here that the dominance of white-transparent colors for fibers indicates they were subjected to longer aging (hence long-range transport) with respect to the colored fragment? It is not clear how you arrive to your conclusion, it will be useful if you could elaborate more explicitly on this point.

Line 399: I would suggest to explicitly say what the ANOVA analysis is, at least the first time you mention it.

Response to Reviewers' Comments

Dear Reviewers,

We would like to thank you for the thoughtful and helpful comments on our manuscript. We have carefully addressed all comments and revised the manuscript accordingly. Below, we provide detailed responses to all your comments. The reviewers' comments are presented in ***bold Times New Roman italics*** font, and our response to the science-specific comments is written in normal font following each of the individual comments. Revisions to the manuscript are included directly in the response to reviewers document, where additions are marked in blue and deletions are ~~struck out in red~~. In addition, all the line numbers in the response to reviewer documents refers to the “revised manuscript with marked changes”.

We feel it is important to note that the reanalysis of 100% of the filters inside the MBL has now been performed and the results are included in the revised version of the manuscript, and the new data are in line with the previous analysis and support the main results. Specifically, our findings demonstrate that the migration of microplastic fibers is more efficient than microplastic fragments, as supported by the hemispheric measurements of airborne microplastics and predictive calculations from back-trajectory measurements (which have also been further developed through the revision process). This work highlights the migration of microplastics from land to the Southern Ocean, which will be of great interest to those working in the field.

Best regards

Guitao Shi on behalf of all co-authors

Reviewer #1 Comment 1. *I read the manuscript “Long-range Atmospheric Transport of Microplastics across the Southern Hemisphere” dealing with the atmospheric transport of microplastics to remote regions. The authors present a hemispheric-scale analysis of airborne microplastics taken during a cruise path. It is true that the transport efficiency of microplastics in the atmosphere is still poorly known, not only in the Southern Hemisphere, but globally, and the data presented are interesting. However my feeling is that they are more routine than a real breakthrough: the results showed that fibers were more efficiently transported, which is a known fact. Besides, the discussion provided by the authors and the interpretation of data present some shortcomings. In include details below.*

Response:

Thank you so much for pointing out the strengths of our study as well as offering suggestions for improving the manuscript’s discussion and significance. The primary contributions of this study are as follows: (1) it is the first to report hemispheric transport of airborne microplastics, even though it utilizes well-established monitoring and characterization methods, (2) our findings demonstrate that the migration of microplastic fibers is more efficient than microplastic fragments, as supported by both practical measurements and predictive calculations, and (3) we present evidence of the migration of microplastics from land to the Southern Ocean, which is derived from our analysis of the trends in the weathering indices of certain types of microplastic samples. To make these unique aspects of our study more apparent to the reader, we have supplemented our initial submission with more in-depth discussions and improved the data interpretation quality carefully, as suggested by all reviewers.

Reviewer #1 Comment 2. *I think there is a factual error in line 56 concerning the cite of D. Allen. The size range is not 0.2-5000 μm , but 20 μm -5 mm: “However, this marine sampling comprises particles collected predominantly in the range of 20 μm -5mm”.*

Response:

Thank you very much for pointing this out. We have revised the expression in the manuscript:

“For instance, airborne micro(nano)plastics (MnPs) monitored by current technology have a typical size range of ~~0.2-5000 μm~~ ~~###~~ 20 μm -5 mm, making them noticeably larger than the ~~typical~~-aerosols typically found in the atmosphere (e.g., sea salts) ~~in the atmosphere.~~” (Lines 53-57)

Reviewer #1 Comment 3. *0.2 μm is already in the nanoplastic range and mentioning this size range is risky. Currently, there are no techniques for detecting nanoplastics in environmental samples except in very specific cases. It is true that some authors reported high concentrations of nanoplastics, even much higher than the concentration of most pollutants in wastewater, which makes them unbelievable. I know this is the way to gain headlines, but extraordinary claims require extraordinary evidences.*

Response:

Thank you for expressing your concerns on this point. We completely agree with the perspectives you have shared. It is quite true that currently the detection of nanoplastics in environmental samples is limited, with only a few specific cases reported where high concentrations were observed. Therefore, we revised the expression as follows.

“For instance, airborne micro(~~nano~~)plastics (~~MnPs~~) monitored by current technology have a typical size range of ~~0.2–5000~~ ~~µm~~ 20 µm–5 mm, making them noticeably larger than the ~~typical~~ aerosols typically found in the atmosphere (e.g., sea salts) ~~in the atmosphere.~~” (Lines 53-57)

In our study, only microplastics above 20 µm were determined using micro-FTIR. This point has been clarified in the Methods section of the paper in order to avoid making extraordinary claims.

Reviewer #1 Comment 4. *The authors claim (several times) that they used a unified and standardized monitoring method, but this was because they were the results of a single expedition. It would have been strange that they didn’t unify methods with themselves. Comparing data from different studies is not a weakness (line 82), but a strength because of result cross-checking.*

Response:

With this statement we meant that comparing studies performed in different regions to understand the geographic distribution of airborne microplastics is challenging because all groups carry out microplastic sampling and analysis differently. The data analyzed in this study were obtained from a single expedition, and a large number of representative sampling points (26 points, spanning from latitude ~30°N to ~70°S) were assessed in the same manner.

To the best of our knowledge, this sample size, and particularly the latitudes which are covered, is larger than any previously reported voyages aiming to quantify atmospheric microplastics. We agree that comparing our results with other studies is beneficial despite the lack of a widely adopted sampling and analysis protocol. Consequently, we have included a new figure (Figure S3) and a new table (Table S3) regarding the atmospheric microplastics measured between previous studies and the current study. To make a reasonable comparison, especially in light of the reviewer’s previous comments, references included microplastic studies and excluded nanoplastic studies.

We have also indicated the analytical method(s) which were used in each case, including the method size range, as well as the environment in which the samples were taken. We found that the airborne MPs concentrations detected in this study are not significantly different from previous studies sampled over the open ocean or coasts, but significantly lower than those collected over the land by orders of magnitude ($p < 0.05$).

“The airborne MPs concentrations measured in this study ($0.035 \pm 0.009 \text{ n m}^{-3}$) were not significantly different from previous studies sampled over the open ocean or coasts ($2.12 \pm 3.84 \text{ n m}^{-3}$; $p=0.243$), but significantly lower than those collected over the land by orders of magnitude ($24.77 \pm 40.61 \text{ n m}^{-3}$; $p=0.040$) (Table S3, Figure S3).” (Lines 121-125)

“The analytical approach we used in this study is in line with approaches which are typically used for collection and characterization of airborne microplastics, despite there being no standardized methods across studies.” (Lines 578-580)

Table S3 Summary of airborne microplastics concentrations collected with air samplers.

Microplastic air concentrations sampled over the ocean or coasts					
Location	Average microplastic counts ^a	Analysis method	Environment	Size range ^b	Reference

From Pacific Ocean to Southern Ocean	0.039, 0.032, 0.041, 0.048, 0.035, 0.038, 0.032 MP/m ³ (offshore and pelagic areas)	μFTIR	Offshore and open ocean air	20 μm -5 mm	This study
Atlantic coast, France	2.9, 9.6 MP/m ³ (on shore and off shore)	μRaman	Onshore and offshore air	2.5 μm-300 μm	(Allen et al. 2020)
Atlantic Ocean	0.0112 MP/m ³	μRaman	Offshore air	5 μm-5 mm	(Trainic et al. 2020)
South China Sea	0.39 MP/100m ³	μFTIR	Offshore air	20 μm -1 mm	(Wang et al. 2021)
Western Pacific Ocean	0.13, 0.01 MP/m ³ (coastal and pelagic areas)	μFTIR	Onshore and offshore air	20 μm -2 mm	(Liu et al. 2019b)

Microplastic air concentrations sampled over the land

Location	Average microplastic counts ^a	Analysis method	Environment	Size range ^a	Reference
Safat, Kuwait	8.9, 14.6, 4.15, 14.3 MP/m ³	μRaman	City air	1 μm-5 mm	(Uddin et al. 2022)
Weser River catchment, Germany	121, 37, 115 MP/m ³	Raman	Rural and urban air	4 μm-5 mm	(Kernchen et al. 2022)
Pic du Midi, France	0.23 MP/m ³	μRaman	Rural air	5 μm-163 μm	(Allen et al. 2021)
Shanghai, China	1.42 MP/m ³	μFTIR	City air	20 μm-9.55 mm	(Liu et al. 2019a)
Cal State University, USA (outdoor)	7.9 MP/m ³	μRaman, FTIR	City air	20 μm ->3 mm	(Zhang et al. 2020)
Madrid, Spain	1.5, 3.2, 3.7, 13.9 MP/m ³ (from rural to urban)	μFTIR	Rural and urban air	25 μm-5 mm	(González-Pleiter et al. 2021)

^a In some rows, multiple numbers representing the average concentrations of various sampling regions are presented for a single study.

^b References include microplastics studies but exclude nanoplastics studies.

Figure S3 Comparison of atmospheric MPs concentrations between previous studies and this study, with further detailed information in Table S3. * indicates a statistical difference between the two groups according to independent t tests ($p < 0.05$). In reporting the results for this study, we calculated and plotted seven average values which correspond to the latitudes of 0-15°N, 15-30°N, 0-15°S, 15-30°S, 30-45°S, 45-60°S, and 60-75°S, which are presented in the graph as one data point. Lines that cross the violin plots are median values.

Reviewer #1 Comment 5. Lines 151-152: Reference 14 does not refer to fibres.

Response:

Thank you very much for bringing up this point. It is true that the authors mainly used microplastic standards (sub-2 μ m, PS spheres; super-2 μ m, PE and PP pellets) as their main model materials. They calculated the bottom-up emission estimates of oceanic microplastics (0.3-70 μ m) were 24 (~1–47) quintillion pieces or 773 (~30–1515) tons. This study (Yang et al. 2022) did mention and discuss fibers, but only in the Supporting Information in the section “Emission of plastic fibers” on page S4. In that test, they found the emissions of 100 μ m long PET, PVA, and PP fibers with an average diameter of 25 μ m, 31 μ m, and 40 μ m, respectively. The results of this experiment were that “these fibers were not observed in the collected sea spray aerosols”. However, as that was a small-scale experiment with limited polymer chemistries and lengths of MPs fibers, and was conducted with small-scale lab equipment, we revised the sentence as follows.

“While it is plausible to consider that fibers can be emitted from the ocean surface and be released to the atmosphere, in a small scale-laboratory study investigating a restricted range of polymer types and fiber lengths (diameters ranging from 25 – 40 μ m with an average length of 100 μ m), fibers were not emitted in significant quantities compared to spheres and fragments (Yang et al. 2022). This may be because fibers have increased surface tension compared to other MPs morphologies. Hence, we assumed that long-range atmospheric transport was responsible for most of the fibers collected in atmospheric samples in this study” (Lines 190-197)

Reviewer #1 Comment 6. Concerning chemical characterization, regenerated cellulose is very difficult to distinguish from cotton or other cellulosic materials using spectroscopic methods. As rayon dominates the composition of the samples taken in this work this is not a minor issue as natural fibres have been shown to dominate airborne fibres in other studies. The

authors should discuss the details of the spectra in the region 3000-3800 cm⁻¹ that support the attribution to rayon of a high number of fibres.

Response:

Thanks very much for this suggestion to point out these challenges to future readers. It is true that rayon (a man-made fiber) is challenging to distinguish from cotton or other cellulosic materials using spectroscopic methods. As this was also an issue that occurred during our field investigation and subsequent laboratory analysis, our group has developed a method to accurately distinguish rayon from cotton (Cai et al. 2019). The characteristic peak at 1105 cm⁻¹ is very distinct for rayon, which can be applied to distinguish man-made rayon from natural cotton. Therefore, this is an important feature for polymer identification in this study.

Furthermore, we have also found another characteristic peak for distinguishing rayon of the broad and featureless shapes of the band at 3330 cm⁻¹ according to (Comnea-Stancu et al. 2017). This band is the characteristic profile of the O–H stretching band. The shape of this band is rather broad and featureless for man-made fibers, whereas it exhibits a distinct maximum at 3330 cm⁻¹ for natural fibers.

Therefore, we can confidently distinguish rayon and cotton in this study by introducing the two methods described above. We have revised the manuscript as follows in the Methods section.

“Considering that the man-made fiber rayon is challenging to distinguish from cotton or other cellulosic materials, we have adopted two methods for its accurate identification. First, the characteristic peak at 1105 cm⁻¹ is very distinct for rayon, and we have embedded this information into the spectra matching library according to our previous research (Cai et al. 2019). Second, another characteristic band at 3330 cm⁻¹ of the O–H stretching can also be utilized to distinguish rayon, where its band is rather broad and featureless, whereas oppositely it exhibits a distinct maximum at 3330 cm⁻¹ for natural fibers (Comnea-Stancu et al. 2017).” (Lines 597-604)

Reviewer #1 Comment 7. The discussion in the paragraph starting in line 198 (and Ref. 26) refers to equivalent diameters, which is a concept absent from the current manuscript. Beside, ImageJ cannot obtain surface area. The authors probably mean projected area.

Response:

Thank you very much for your pertinent suggestions. Indeed, our use of the term “equivalent diameters” was not explained in detail in the previous version of the manuscript. As the reviewer correctly points out, ImageJ can only present the mean projected area for fragments and fibers. Therefore, after obtaining the width, length, and the calculated height parameters, we estimated the surface area and volume (mass) values of fibers and fragments with different models. For fibers, the cylinder model was applied according to Simon et al. (2018); for fragments, the column model was applied according to a previous study (Koelmans et al. 2020) together with an approximation by assuming the length-to-width ratio equates to the width-to-height ratio ($L/W=W/H$) (Mintenig et al. 2020, Simon et al. 2018).

For the fibers, we measured the projected width and length of particles utilizing ImageJ. The width was equivalent to the diameter of the bottom surface of a cylinder, the length was equivalent to the height of the cylinder, and we calculated the surface area according to Eq S1. For the fragments, we measured the projected length and width of particles utilizing ImageJ, projected the height using $L/W=W/H$, and calculated the surface area according to Eq S2.

$$SA_{\text{fiber}} = 2\pi R \cdot L + \pi R^2/2 \quad (\text{Eq. S1})$$

$$SA_{\text{fragment}} = 4LW + 2W^2 \quad (\text{Eq. S2})$$

where SA represents surface area, R is the diameter and L is the length obtained from ImageJ for fiber particles, W is the width and L is the length obtained from ImageJ for fragment particles.

Collectively, after obtaining the width and length parameters, we estimated the surface area of each particle (Figures 4 and S12). The explanations detailed above were included in the revised Supporting Information (Text S3).

“We used the cylinder model (Simon et al. 2018) and the column model (Koelmans et al. 2020) to estimate the surface area and volume (mass) values of fibers and fragments (details see Text S3). Additionally, an approximation by assuming the length-to-width ratio equates to the width-to-height ratio ($L/W=W/H$) was also applied for fragments (Mintenig et al. 2020, Simon et al. 2018). However, it is important to note that this simplified approach to suggest surface area and volume (mass) likely represents a low estimate, as it does not take into account additional surface roughness and cracks which are likely to be present on environmental microplastics, subsequently increasing the actual surface area.” (Lines 631-638)

Reviewer #1 Comment 8. What is the pore size of glass fibre filters and how the authors measured the flow rate? Didn't the flow rate change during sampling? This is the reason why sampled volumes are generally much lower than those reported in this work. Sampling details are not sufficiently detailed in the manuscript.

Response:

Thanks very much for the comments. We have supplemented the details about pore size of the glass fiber filters (2.5 μm), flow rate (1.2 $\text{m}^3 \text{min}^{-1}$), and sampling volume which are now included in the revised manuscript. Additional information on how the filters were placed, removed from the sampler and stored prior to analyses was also included.

“Atmospheric particles were collected onto Whatman quartz fiber filters (~~Whatman QM-A, 8×10 in~~, 20.3 cm × 25.4 cm, with the pore size of 2.5 μm). At each site, the membranes were prebaked at ~500 °C for > 6 h to ensure the filters were free of polymer particles.” (Lines 528-531)

“For mounting the filters, the buckles of the upper placing plate were unscrewed after opening the upper cover of the HVAS, and then a filter membrane was laid flush on the lower plate. Then the buckles were re-tightened, and the upper cover of the HVAS was closed. The entire process was generally completed within about 30 s, reducing the chance of airborne contamination. Note that the filter has one rough surface and one smooth surface, and atmospheric particles were always collected on the rough surface. Typically, the pump of the high-volume air sampler (TISCH Environmental, USA) was running at a constant flow rate, 1.2 $\text{m}^3 \text{min}^{-1}$, and for a sampling duration of 48 h which would consequently lead to sampling a volume of 3456 m^3 . The flow rate of the pump was calibrated when manufactured in the factory. ~~The airflow rate of the HVAS was relatively constant, at 1.2 $\text{m}^3 \text{min}^{-1}$, and a sampling duration of 2-3 days for each sample led to typical sampling air volumes of 3000-4000 m^3 . Typically, a~~ Atmospheric particles along the 2-3 days' cruise path, covering approximately 2-4 degrees of latitude, were collected on one filter which was considered one aggregate sample. A wind direction sensor was employed to control the HVAS to avoid potential contamination from the vessel, so consequently only air masses from a sector ~120° left and right of the central line of the vessels' path was sampled. After sampling, the upper cover of the HVAS was opened and buckles of the upper placing plate were unscrewed, then the filter was removed by pre-cleaned stainless tweezers from the sampler.” (Lines 531-550)

Reviewer #1 Comment 9. Carbonyl index is sometimes used as a measurement of photochemical ageing, but it is cumbersome to assume that it will systematically increase during long-range transport, which can last hours or a few days, depending on the atmospheric circulation.

Besides Fig. 5 is non-conclusive and does not support authors' opinion. Additionally, I don't see any peak in the 1735 cm^{-1} region in the rayon spectrum of Fig. 2, where C=O stretching band should be visible.

Response:

Thanks very much for pointing this out. It is true that the carbonyl index will not linearly increase with latitudes alteration during long-range transport. Therefore, when analyzing environmental samples, a large sample size is important to indicate a general trend of the carbonyl index. Moreover, the carbonyl index is a better indicator of weathering for polymers whose initial polymer structure does not contain C=O groups. For instance, polyester contains carbonyl groups and consequently any newly formed carbonyl signals attributed to polymer aging may not be easily distinguishable from the background (Li et al. 2023). Therefore, the carbonyl index alteration has not been observed for polyester fibers after UV weathering (Pinlova and Nowack 2023). For the two reasons mentioned above, we chose to focus on rayon for a cross-latitude carbonyl index alteration analysis, which had a large number of microplastics consistently recovered in samples across all latitudes and the material does not contain C=O in its pristine form.

We agree that the previous Figure 5 was indeed non-conclusive because we only obtained a general trend with a limited number of microplastics. However, we did find that the degree of rayon aging at low latitudes was significantly lower than that at higher latitudes. The rayon particle shown in Figure 2 is an example collected at a low latitude. As rayon itself does not have a C=O group in its pristine form, the peak at 1735 cm^{-1} is not quantifiable for this fiber. We have further supplemented the diagram of rayons' spectra comparison between low and high latitudes, where the change in the C=O position (at 1735 cm^{-1}) is more clearly shown (Figure S15).

Changes to the Supporting Information:

Figure S15 Rayon micro-FTIR spectra comparison between samples collected from low and high latitudes. As the carbonyl

index is a better indicator of weathering for polymers whose initial polymer structure does not contain C=O groups, and it would not linearly increase with latitudes alteration during long-range transport, we chose to focus on rayon for a cross-latitude carbonyl index alteration analysis, which was frequently identified in samples collected. Furthermore, rayon does not contain C=O groups in pristine form. The C=O peak (at 1735 cm^{-1} , blue lines) is more clearly shown for rayon collected at higher latitudes, and the area under 1800 - 1670 cm^{-1} (red line intervals) was deemed as the absorbance area (A1) of the carbonyl (C=O) group, and the area under 1500 - 1390 cm^{-1} was deemed as the absorbance area (A2) of the reference (CH₂) group in Eq.3.

Reviewer #1 Comment 10. In sum, although the authors presented interesting results, neither their novelty nor the depth of discussion is sufficient to gran publication in a top journal like NC.

Response:

We thank the reviewer for the critiques provided on several aspects of this study, which allowed us to improve our manuscript. We have further highlighted the novelty in the revised manuscript, in addition to providing clarity to the questions posed by the reviewer. A summary of the novelty is detailed in the response to this reviewers' first comment, and we hope to have satisfied the other scientific queries by providing additional information and contextualization. We believe the paper is greatly strengthened with this additional information, more in-depth discussion, and clarity on data quality and interpretation.

End of responses to Reviewer#1

Reviewer #2 (Remarks to the Author):

Reviewer #2 Comment 1. Dear authors, you present noteworthy results of the presence of MPs in the atmosphere across the Southern hemisphere. your work is of significance to the field of atmospheric MP research. Some information in the methods and the subsequent use of the acquired data does not always support the conclusions and claims, with additional input and revision needed? Some flaws in the data analysis, interpretation and conclusions are pointed out below in more detail, requiring revision.

Response:

Thank you for your positive comments! According to your detailed suggestions, we have added more information to the methods and substantially enhanced the interpretation and discussion of the results.

Reviewer #2 Comment 2. Line 104: Please also see comment on method chapter, but the described use of only 30% of 1/4th of the filter results in 1.4 and less MPs per sample. The reanalysis of 100% of the 1/4 of the filter is strongly advised for all samples inside the MBL.

Response:

Thank you very much for your advice. The reanalysis of 100% of the ¼ of the filters inside the MBL has now been performed and the results are included in the revised version of the manuscript. During manuscript revision, we reanalyzed 14 filter samples where we had previously only measured 30% of the randomly selected particles on the filter. Comparing the results of both analyses, the concentrations of both MPs and non-plastic particles between sub-sampling (1st analysis) and the full analysis (2nd analysis) were similar. Further explanation has been added (see Text S2) and the figures have subsequently been updated.

Text S2 Atmospheric particles concentrations comparison between the sub-sampling and full analysis

Atmospheric particles were collected onto Whatman quartz fiber filters (20.3 cm × 25.4 cm) at each site. Typically, for a sampling duration of 48 h, this would consequently lead to sampling a volume of 3456 m³ on each filter. For sample analysis, one fourth of each filter was used for particles characterization and quantification according to a previous airborne particulate study (Moch et al. 2020). Two morphologies were included here, namely fibers and fragments, and two rounds of analysis were conducted, as detailed below.

1st analysis:

As the fibers number concentrations decreased for more southerly latitudes, we found that 14 samples in the more northerly sample locations had too many fibers on one fourth of the filters to easily measure (number of fibers exceeded one hundred). To facilitate the identification and quantitation, all fibers were observed and collected under the microscope and the total numbers were counted. Then, we randomly selected 30% of the fibers to identify their composition chemistries. Therefore, of the 26 sampling sites, we identified all fibers for 12 sampling sites (i.e., southern sites) and 30% of the fibers for 14 sampling sites (i.e., northern sites), and adjusted the final fiber number concentrations and chemical identification reported accordingly during the 1st analysis. Fragments in all 26 samples were counted and chemical compositions were identified across the entire expedition.

2nd analysis:

For method verification, we used the backup samples (another one fourth of the filters) during the 2nd analysis for a full scan to prove the robustness of this sub-sampling approach (i.e., only using 30% of the fibers in the 1st analysis) by comparing these values with the entire analysis (100% of fibers analyzed in the 2nd analysis). The impacts of sub-sampling a filter versus identifying the entire sample did not have large discrepancies in this instance (Table S6, Figure S17). Here, the fiber concentrations obtained between sub-sampling and the full analysis were similar, when scaled for the proportion of the filter which was analyzed. Beyond our specific study, this may be useful for other researchers in the future to assess whether a full scan needs to be performed or only a sub-section analysis is sufficient in situations where a large number of microplastics are recovered. However, we appreciate that by nature microplastics contamination can be heterogeneous, and so an assessment of the goodness of fit of subsampling should be considered in any study which does not measure the entire particle distribution in a sample.

During both the 1st and the 2nd analysis, all fragments (100%) on filters were identified, and a comparison of the fragments concentrations indicates that there is no significant difference between the two measurements for MPs fragments and non-plastic fragments according to *t*-tests (Table S7, Figure S18).

Table S6 Comparison of MPs and non-plastic fibers concentrations between an entire analysis (100% particles were fully scanned) and a sub-sampling analysis (30% randomly selected particles on the filter) for 14 samples.

Latitude (°)	MPs Fibers ^a (n·m ⁻³)	MPs Fibers ^b (n·m ⁻³)	Non-plastic Fibers ^a (n·m ⁻³)	Non-plastic Fibers ^b (n·m ⁻³)
-64.39	0.0237	0.0284	0.0671	0.0675
-64.74	0.0344	0.0309	0.0573	0.0378
-65.17	0.0235	0.0317	0.1135	0.1045
-65.27	0.0228	0.0342	0.1065	0.0867
-44.41	0.0263	0.0259	0.0788	0.0788
-34.27	0.0428	0.0443	0.1128	0.1038
-24.05	0.0335	0.0413	0.0781	0.0737
-13.66	0.0321	0.0301	0.0763	0.0458
-4.71	0.035	0.0303	0.0738	0.0699
5.12	0.0277	0.0291	0.1039	0.1028
12.79	0.0177	0.0146	0.0886	0.0864
18.67	0.014	0.0210	0.0842	0.0431
24.97	0.0261	0.0257	0.0819	0.0760
30.20	0.0379	0.0391	0.1221	0.1136

^a indicates data where 30% of fibers were measured on ¼ filter and scaled, shown in black values.

^b indicates data where 100% of fibers were measured on ¼ filter, shown in blue values.

Figure S17 MPs and non-plastic particle concentrations comparison between the sub-sampling analysis (1st analysis, 30% of fibers randomly selected and scaled) compared to measurement of all fibers in the sample (2nd analysis, 100% particles quantified) for 14 northerly samples across the campaign. (A) MPs fibers; (B) Non-plastic fibers. ns: indicates there was no statistically significant difference between the two groups ($p > 0.05$) according to t -tests.

Table S7 Comparison of MPs and non-plastic fragments concentrations between the 1st and the 2nd analysis. We made replicate analysis on two different $\frac{1}{4}$ of the whole filter, and the fragments concentrations results do not show statistically significant differences according to t -tests.

Latitude (°)	MPs Fragments ^a (n·m ⁻³)	MPs Fragments ^b (n·m ⁻³)
-64.3882	0.0024	0.0059
-64.7367	0.0023	0.0046
-65.1668	0.0035	0.0047
-65.2672	0.0046	0.0103
-44.4146	0.0045	0.0056
-34.2737	0.0035	0.0035
-24.0535	0.0067	0.0067
-13.657	0.0084	0.0121
-4.7144	0.0105	0.0105
5.1183	0.0083	0.0114
12.7883	0.0106	0.0080
18.6733	0.0063	0.0105
24.9708	0.0078	0.0123
30.2017	0.0038	0.0088

- a** results for the 1st membrane analysis (1/4 of the whole filter (20.3 × 25.4 cm)), shown in black values;
b results for the 2nd membrane analysis (1/4 of the whole filter (20.3 × 25.4 cm)), shown in blue values.

Figure S18 MPs and non-plastic fragments concentrations between the the 1st and the 2nd one-fourth filters. (A) MPs fragments; (B) Non-plastic fragments. ns: indicates there is no significantly statistical difference between the two groups ($p > 0.05$) according to t -tests.

Reviewer #2 Comment 3. Line 118: *The use of a density of 1.4 g/cm³ covers only some polymers, with the most prevalent ones in terms of global production and use are characterized by much lower densities below or close to 1 (PE, PP, PS). State alternative sedimentation rates for these polymers as well.*

Response:

Thanks very much for the kind reminder. The alternative sedimentation rate for MPs with lower densities are now also provided as follows.

“Similarly, the lower ρ_p corresponds to lower U ; e.g., polyethylene (0.91-0.94 g·cm⁻³) and polystyrene (0.96-1.05 g·cm⁻³), their estimated sedimentation speed is approximately 1.0-1.2 m·s⁻¹.” This point was included in the revised version. (Lines 153-155)

Furthermore, we also calculated the updraft wind speed for MPs with lower densities (i.e., 0.6, 0.8, and 1.0 g·cm⁻³) in the figure. We have subsequently moved this figure to the main text as Figure 3 in the revised version.

Figure 3 Changes in the vertical component of the updraft wind speed which can carry microplastics with different lengths and densities. L is the length of a particle and $2r$ is its diameter or width; and L is defined to be always larger than $2r$. The smaller of the $L/2r$ ratio is, the shape of the particle is closer to a fragment; whereas the larger of the $L/2r$ ratio is, the shape of the particle is closer to a fiber. ρ_p indicates the density of a plastic particle.

Reviewer #2 Comment 4. Line 154 ff

Rayon is a semi-synthetic polymer, consisting of cellulose. The discussion of the ability of the FTIR to distinguish between natural cellulose fibres and rayon is missing. As a consequence, the inclusion of data on rayon should be limited to general descriptions rather than detailed data evaluation and discussion.

Response:

Thanks very much for the comments. We have supplemented the discussion of the ability of the FTIR to distinguish between natural cellulose fibers and rayon.

“Considering that the man-made fiber rayon is challenging to distinguish from cotton or other cellulosic materials, we have adopted two methods for its accurate identification. First, the characteristic peak at 1105 cm^{-1} is very distinct for rayon, and we have embedded this information into the spectra matching library according to our previous research (Cai et al. 2019). Second, another characteristic band at 3330 cm^{-1} of the O–H stretching can also be utilized to distinguish rayon, where its band is rather broad and featureless, whereas oppositely it exhibits a distinct maximum at 3330 cm^{-1} for natural fibers (Comnea-Stancu et al. 2017).” (Lines 597-604)

Moreover, the inclusion of data on rayon has been limited to general descriptions rather than detailed data evaluation and discussions. We have removed the detailed description of the rayon carbonyl index along the latitudes and the original Figure 5 from the main text, and moved them into the Supporting Information.

“We observed a higher CI index for MPs collected from the MBL at higher latitudes compared to mid-low latitudes (Figure S14), and the change of the C=O position (at 1735 cm^{-1}) was clearly shown (Figure S15), suggesting increased weathering during long range transport. ~~We only compared the CI index for rayon MPs as it has the most abundant occurrence for both~~

~~fibers and fragments (Figure 5).” (Lines 259-264)~~

~~“where, A_1 is the absorbance of the carbonyl (C=O) peak for MPs rayon fibers and resin fragments, and A_2 is the absorbance of a reference peak (CH₂) for these MP samples. Detailed area bands were depicted in Text S4 and Figure S15. Here, A_1 area bands used for rayon and resin were of 1700-1570 cm⁻¹ and 1800-1650 cm⁻¹, respectively; and A_2 area bands used for rayon were of 1500-1390 cm⁻¹ and 1500-1350 cm⁻¹, respectively.” (Lines 661-666)~~

Reviewer #2 Comment 5. Line 209 ff

Since the weathering evaluation was only done with rayon with its challenges to be correctly distinguished from natural cellulose, and rayon being the almost only polymer with a considerable number of CH₃- and C=O groups potentially formed, this part is highly speculative and not supported by the available data. Please remove.

Response:

As mentioned in the response to the above *Comment 4* from this reviewer, we have deleted the detailed description of the rayon carbonyl index along the latitudes and the original Figure 5 in the main text and moved them into the Supporting Information. Though we observed the formation of C=O groups of rayon MPs collected at high latitudes, this information was only provided in the Supporting Information (Figure S14, Figure S15, and Text S4). Further discussion on this point has also been included in **Text S4**).

Text S4 Carbonyl index calculation.

It is true that the carbonyl index will not linearly increase with latitudes alteration during long-range transport. Therefore, when analyzing environmental samples, a large sample size is important to indicate a general trend of the carbonyl index. Moreover, the carbonyl index is a better indicator of weathering for polymers whose initial polymer structure does not contain C=O groups. For instance, polyester contains carbonyl groups and consequently any newly formed carbonyl signals attributed to polymer aging may not be easily distinguishable from the background (Li et al. 2023). Therefore, the carbonyl index alteration has not been observed for polyester fibers after UV weathering (Pinlova and Nowack 2023). For the two reasons mentioned above, we chose to focus on rayon for a cross-latitude carbonyl index alteration analysis, which has a large sample number and the material does not contain C=O in its pristine form. We have further supplemented a diagram of rayon's spectra comparison between low and high latitudes (Figure S15), where the C=O peak (at 1735 cm⁻¹, blue lines) was more clearly shown for rayon collected at higher latitudes, and the area under 1800 - 1670 cm⁻¹ (red line intervals) was deemed as the absorbance area (A_1) of the carbonyl (C=O) group, and the area under 1500 - 1390 cm⁻¹ was deemed as the absorbance area (A_2) of the reference (CH₂) group in Eq 3.

Figure S15 Rayon micro-FTIR spectra comparison between samples collected from low (A-B) and high (C-D) latitudes. As the carbonyl index is a better indicator of weathering for polymers whose initial polymer structure does not contain C=O groups, and it would not linearly increase with latitudes alteration during long-range transport, we chose to focus on rayon for a cross-latitude carbonyl index alteration analysis, which was frequently identified in samples collected. Furthermore, rayon does not contain C=O groups in pristine form. The C=O peak (at 1735 cm⁻¹, blue lines) is more clearly shown for rayon collected at higher latitudes, and the area under 1800 - 1670 cm⁻¹ (red line intervals) was deemed as the absorbance area (A1) of the carbonyl (C=O) group, and the area under 1500 - 1390 cm⁻¹ was deemed as the absorbance area (A2) of the reference (CH₂) group in Eq.3.

Reviewer #2 Comment 6. Line 439:

Please add also metric dimensions, since all other sizes are given in metric units.

Response:

Thanks so much for this kind reminder. We have converted the metric dimensions into metric units as follows.

“Atmospheric particles were collected onto Whatman quartz fiber filters (~~Whatman QM-A, 8×10 in~~, 20.3 cm × 25.4 cm, with the pore size of 2.5 μm). At each site, the membranes were prebaked at ~500 °C for > 6 h to ensure the filters were free of polymer particles.” (Lines 528-531)

Reviewer #2 Comment 7. Line 456: State the period of time the field blank was placed in the sampler.

Response:

Four field blank samples (two collected along the cruise path and two collected inland) were prepared from filters mounted in the HVAS with an air pump flow rate set to 0. As the filters can directly have contact with the surrounding air, the period of time was set as “30 s” to avoid potential extra contamination. We have revised the manuscript to include this point.

“Furthermore, four field blank samples (two collected along the cruise path and two collected inland) were prepared from filters mounted in the HVAS with an air pump flow rate set to 0. To avoid the potential contamination from the surrounding

air environment, the period of time for the blank filters was set as “30 s”. For the field blanks, no MPs were found.” (Lines 557-561)

Reviewer #2 Comment 8. Line 461 ff:

Please extend on the information on how the filter was placed, removed from the sampler and stored prior analyses.

Response:

Thanks for the question and the opportunity to further clarify the methods we used in this study. In the revised manuscript, we have provided additional details about the operation process.

“For mounting the filters, the buckles of the upper placing plate were unscrewed after opening the upper cover of the HVAS, and then a filter membrane was laid flush on the lower plate. Then the buckles were re-tightened, and the upper cover of the HVAS was closed. The entire process was generally completed within about 30 s, reducing the chance of airborne contamination. Note that the filter has one rough surface and one smooth surface, and atmospheric particles were always collected on the rough surface. Typically, the pump of the high-volume air sampler (TISCH Environmental, USA) was running at a constant flow rate, $1.2 \text{ m}^3 \text{ min}^{-1}$, and for a sampling duration of 48 h which would consequently lead to sampling a volume of 3456 m^3 . The flow rate of the pump was calibrated when manufactured in the factory. ~~The air flow rate of the HVAS was relatively constant, at $1.2 \text{ m}^3 \text{ min}^{-1}$, and a sampling duration of 2-3 days for each sample led to typical sampling air volumes of $3000-4000 \text{ m}^3$. Typically, a~~ Atmospheric particles along the 2-3 days’ cruise path, covering approximately 2-4 degrees of latitude, were collected on one filter which was considered one aggregate sample. A wind direction sensor was employed to control the HVAS to avoid potential contamination from the vessel, so consequently only air masses from a sector $\sim 120^\circ$ left and right of the central line of the vessels’ path was sampled. After sampling, the upper cover of the HVAS was opened and buckles of the upper placing plate were unscrewed, then the filter was removed by pre-cleaned stainless tweezers from the sampler. ~~After sampling, individual~~ Individual filters were kept separate, folded, wrapped in aluminum foil, placed in zip-loc bags, and stored in the dark under freezing conditions ($-20 \text{ }^\circ\text{C}$) until particle characterization processing began.” (Lines 531-552)

Reviewer #2 Comment 9. Line 493:

Please specify if the analysed MPs represented 30% of the 1/4 of the filter? Can you show that this is representative for the whole filter also from filters from inside the MBL with a low MP count?

Response:

Thank you very much for this kind advice. The reanalysis of 100% particles of the $\frac{1}{4}$ of the filters inside the MBL has now been performed and the results are included in the revised version of the manuscript. During the manuscript revision, we reanalyzed fourteen filter samples where we had previously only measured 30% particles in the filter area.

We used the corresponding backup samples for a full scan to better compare this sub-sampling (30%, as initially done) with an entire analysis (100%, now performed). In the updated figures and tables, we consequently show the impact of sub-sampling a filter versus measuring the entire sample (Table S6, Figure S17). In this case, the number and characteristics of particles which were obtained between sub-sampling and the full analysis were similar, when scaled for the proportion of the filter which was analyzed. Beyond our specific study, this may be useful for other researchers in the future to assess if an

entire filter needs to be analyzed or only a sub-section, since if a sub-section is sufficient, this would consequently save time and effort across the analytical quantification step. We have updated Figure 1 and other relevant figures in the revised manuscript to include the results from both analysis methods, where the same trends for fibers and fragments were observed independent of the extent of filter analysis. Detailed explanation has also been added in the revised manuscript and Text S2.

“For fibers, we found that 14 samples in the more northerly sample locations had too many fibers on one fourth of the filters to easily measure (number of fibers exceeded one hundred). To facilitate the identification and quantification, all fibers were observed and collected under the microscope and total numbers were counted. Then, we randomly selected 30% of the fibers on these 14 filters to identify their composition chemistries. For this method verification, we used the backup samples (another one fourth of the filters) during the 2nd analysis for a full scan to prove the robustness of the sub-sampling approach (i.e., only using 30% of the fibers in the 1st analysis) by comparing these values with the entire analysis (100% of the fibers analyzed in the 2nd analysis) (see Text S2). ~~when the number of fibers in a sample exceeded 100, one third of the randomly selected items were representative of the entire fiber population collected on a filter (according to the results of the first twelve sampling sites examined).~~” (Lines 606-618)

Text S2 Atmospheric particles concentrations comparison between the sub-sampling and full analysis

Atmospheric particles were collected onto Whatman quartz fiber filters (20.3 cm × 25.4 cm) at each site. Typically, for a sampling duration of 48 h, this would consequently lead to sampling a volume of 3456 m³ on each filter. For sample analysis, one fourth of each filter was used for particles characterization and quantification according to a previous airborne particulate study (Moch et al. 2020). Two morphologies were included here, namely fibers and fragments, and two rounds of analysis were conducted, as detailed below.

1st analysis:

As the fibers number concentrations decreased for more southerly latitudes, we found that 14 samples in the more northerly sample locations had too many fibers on one fourth of the filters to easily measure (number of fibers exceeded one hundred). To facilitate the identification and quantitation, all fibers were observed and collected under the microscope and the total numbers were counted. Then, we randomly selected 30% of the fibers to identify their composition chemistries. Therefore, of the 26 sampling sites, we identified all fibers for 12 sampling sites (i.e., southern sites) and 30% of the fibers for 14 sampling sites (i.e., northern sites), and adjusted the final fiber number concentrations and chemical identification reported accordingly during the 1st analysis. Fragments in all 26 samples were counted and chemical compositions were identified across the entire expedition.

2nd analysis:

For method verification, we used the backup samples (another one fourth of the filters) during the 2nd analysis for a full scan to prove the robustness of this sub-sampling approach (i.e., only using 30% of the fibers in the 1st analysis) by comparing these values with the entire analysis (100% of fibers analyzed in the 2nd analysis). The impacts of sub-sampling a filter versus identifying the entire sample did not have large discrepancies in this instance (Table S6, Figure S17). Here, the fiber concentrations obtained between sub-sampling and the full analysis were similar, when scaled for the proportion of the filter which was analyzed. Beyond our specific study, this may be useful for other researchers in the future to assess whether a full scan needs to be performed or only a sub-section analysis is sufficient if situations where a large number of microplastics are recovered. However, we appreciate that by nature microplastics contamination can be heterogeneous, and so an assessment

of the goodness of fit of subsampling should be considered in any study which does not measure the entire particle distribution in a sample.

During both the 1st and the 2nd analysis, all fragments (100%) on filters were identified, and a comparison of the fragments concentrations indicates that there is no significant difference between the two measurements for MPs fragments and non-plastic fragments according to *t*-tests (Table S7, Figure S18).

Table S6 Comparison of MPs and non-plastic fibers concentrations between an entire analysis (100% particles were fully scanned) and a sub-sampling analysis (30% randomly selected particles on the filter) for 14 samples.

Latitude (°)	MPs Fibers ^a (n·m ⁻³)	MPs Fibers ^b (n·m ⁻³)	Non-plastic Fibers ^a (n·m ⁻³)	Non-plastic Fibers ^b (n·m ⁻³)
-64.39	0.0237	0.0284	0.0671	0.0675
-64.74	0.0344	0.0309	0.0573	0.0378
-65.17	0.0235	0.0317	0.1135	0.1045
-65.27	0.0228	0.0342	0.1065	0.0867
-44.41	0.0263	0.0259	0.0788	0.0788
-34.27	0.0428	0.0443	0.1128	0.1038
-24.05	0.0335	0.0413	0.0781	0.0737
-13.66	0.0321	0.0301	0.0763	0.0458
-4.71	0.035	0.0303	0.0738	0.0699
5.12	0.0277	0.0291	0.1039	0.1028
12.79	0.0177	0.0146	0.0886	0.0864
18.67	0.014	0.0210	0.0842	0.0431
24.97	0.0261	0.0257	0.0819	0.0760
30.20	0.0379	0.0391	0.1221	0.1136

^a indicates data where 30% of fibers were measured on ¼ filter and scaled, shown in black values.

^b indicates data where 100% of fibers were measured on ¼ filter, shown in blue values.

A. MPs fibers

B. Non-plastic fibers

Figure S17 MPs and non-plastic particle concentrations comparison between the sub-sampling analysis (1st analysis, 30% of fibers randomly selected and scaled) compared to measurement of all fibers in the sample (2nd analysis, 100% particles quantified) for 14 northerly samples across the campaign. (A) MPs fibers; (B) Non-plastic fibers. ns: indicates there was no statistically significant difference between the two groups ($p > 0.05$) according to t -tests.

Table S7 Comparison of MPs and non-plastic fragments concentrations between the 1st and the 2nd analysis. We made replicate analysis on two different $\frac{1}{4}$ of the whole filter, and the fragments concentrations results do not show statistically significant differences according to t -tests.

Latitude (°)	MPs Fragments ^a (n·m ⁻³)	MPs Fragments ^b (n·m ⁻³)
-64.3882	0.0024	0.0059
-64.7367	0.0023	0.0046
-65.1668	0.0035	0.0047
-65.2672	0.0046	0.0103
-44.4146	0.0045	0.0056
-34.2737	0.0035	0.0035
-24.0535	0.0067	0.0067
-13.657	0.0084	0.0121
-4.7144	0.0105	0.0105
5.1183	0.0083	0.0114
12.7883	0.0106	0.0080
18.6733	0.0063	0.0105
24.9708	0.0078	0.0123
30.2017	0.0038	0.0088

- a** results for the 1st membrane analysis (1/4 of the whole filter (20.3 × 25.4 cm)), shown in black values;
b results for the 2nd membrane analysis (1/4 of the whole filter (20.3 × 25.4 cm)), shown in blue values.

Figure S18 MPs and non-plastic fragments concentrations between the the 1st and the 2nd one-fourth filters. (A) MPs fragments; (B) Non-plastic fragments. ns: indicates there is no significantly statistical difference between the two groups ($p > 0.05$) according to t -tests.

Reviewer #2 Comment 10. Line 502:

What about the fieldblanks? these are the important blanks that should be used in the QA/QC of the data evaluation. Please add information on the fieldblank MP content and how the data was used to blank correct the field data.

Response:

Thanks very much for the valuable question and suggestion. These field blanks are very important for the QA/QC of the data evaluation. We found no MP (> 20 μm) in all of the field blanks. This is because our clean laboratory environment (observation and identification was conducted in iron-walled labs equipped with air purification systems) and no sample digestion, density flotation, and membrane separation steps were included in our study. As no MPs were detected in the field blank, therefore the MPs results did not need correction. Furthermore, the detailed information about how the field blanks were performed have been described in the main text below.

“Furthermore, four field blank samples (two collected along the cruise path and two collected inland) were prepared from filters mounted in the HVAS with an air pump flow rate set to 0. To avoid the potential contamination from the surrounding air environment, the period of time for the blank filters was set as “30 s”. For the field blanks, no MPs were found. The sampling protocols were the same as those described above in terms of sampling duration, filter mounting, collection, transport, observation, and measurements.” (Lines 557-563)

Reviewer #2 Comment 11. Line 516.

A reference for this method is missing. Was this method tested with spiked samples with known weight? Which mean

density was used here?

Response:

Thanks very much for the kind reminder. We have added the references for both fiber mass and fragment mass calculations and their corresponding references as follows.

“We used the cylinder model (Simon et al. 2018) and the column model (Koelmans et al. 2020) to estimate the surface area and volume (mass) values of fibers and fragments (details see Text S3). Additionally, an approximation by assuming the length-to-width ratio equates to the width-to-height ratio ($L/W=W/H$) was also applied for fragments (Mintenig et al. 2020, Simon et al. 2018). However, it is important to note that this simplified approach to suggest surface area and volume (mass) likely represents a low estimate, as it does not take into account additional surface roughness and cracks which are likely to be present on environmental microplastics, subsequently increasing the actual surface area.” (Lines 631-638)

For the microplastic fibers, we measured the projected width and length of particles utilizing ImageJ. The width was equivalent to the diameter of the bottom surface of a cylinder, the length was equivalent to the length of the cylinder, and calculated the mass according to Equation 1. For the microplastic fragments, we measured the projected length and width, utilizing ImageJ directly, projected the height using $L/W=W/H$, and calculated the mass concentration (MC) and surface area (SA) according to Equation 2.

$$MC_{\text{fiber}} = \sum_{k=1}^n (1-f) \cdot (R_k^2 \cdot L_k) \pi \rho / v_i \quad (\text{Eq. 1})$$

$$MC_{\text{fragment}} = \sum_{k=1}^n S_k \cdot (W_k^2 / L_k) \rho / v_i \quad (\text{Eq. 2})$$

$$SA_{\text{fiber}} = 2\pi R \cdot L + \pi R^2 / 2 \quad (\text{Eq. S1})$$

$$SA_{\text{fragment}} = 4LW + 2W^2 \quad (\text{Eq. S2})$$

where n is in the total number of fibers in the sample, i ; v_i is the sampled air volume, f is the fiber void fraction (40%)(Simon et al. 2018), because airborne fibers become looser than their original states (Figure S16), and the value of $f=0.4$ was used here. R and L were the fiber diameter and length as calculated by the Image J software according to the top-view projection images, respectively. ρ is the average density of primary fibers collected in this survey (Table S4). The above-detailed explanation and calculation was included in the revised Supporting Information (Text S3).

The density used here was the average density of primary fibers and primary fragments collected in this survey. For instance, rayon and polyester were the dominant MPs fibers detected, and rayon and epoxy resin were the dominant MPs fragments detected. Therefore, the average densities for MPs fibers and MPs fragments were calculated based on these primary MPs densities and their composition percentages of 1471 kg/m³ and 1404 kg/m³, respectively. Further, we explained the density selection more clearly in the revised manuscript as follows.

“ ρ is the ~~average density for fragments~~ average density of primary fragments collected in this survey (Table S4).” (Lines 644-646)

Reviewer #2 Comment 12. Line 523:

A reference for this method is missing. How was that method verified?. State that this a method only applicable for rayon and other polymers forming C=O from CH₃- groups.

Response:

Thank you very much for your valuable suggestion! It is true that this method is applicable for rayon but probably not suitable for other polymers and plastics, such as polyester, where the pristine polymer already contains C=O groups. We have added relevant literature that supports this, as explained in further detail below.

First, rayon (a man-made cellulose) is challenging to distinguish from cotton or other cellulosic materials using spectroscopic method. As this was also an issue that occurred during our field investigation, our group has developed a method to accurately distinguish rayon from cotton (Cai et al. 2019). The characteristic peak at 1105 cm^{-1} is very distinct for rayon, which can be applied to distinguish man-made rayon from natural cotton as well. Therefore, this is an important feature for polymer identification in this study, and we have embedded this information into the spectra matching library.

Second, it is true that the carbonyl index will not linearly increase with latitudes alteration during long-range transport. Therefore, when analyzing environmental samples, a large sample size is important to indicate a general trend of the carbonyl index. Moreover, the carbonyl index is a better indicator of weathering for polymers whose initial polymer structure does not contain C=O groups. For instance, polyester contains carbonyl groups and consequently any newly formed carbonyl signals attributed to polymer aging may not be easily distinguishable from the background (Li et al. 2023). Therefore, the carbonyl index alteration has not been observed for polyester fibers after UV weathering (Pinlova and Nowack 2023). For the two reasons mentioned above, we chose to focus on rayon for a cross-latitude carbonyl index alteration analysis, which has a large sample number and the material does not contain C=O in its pristine form. We have further supplemented a diagram of rayons' spectra comparison between low and high latitudes, where the change of the C=O position (at 1735 cm^{-1}) is more clearly shown (Figure S15, Text S1). We also added an explanation in the revised manuscript as follows.

“We chose rayon for a cross-latitude carbonyl index alteration analysis, because rayon (man-made cellulose) which has a large detection rate, does not contain C=O itself, and has been verified to be photochemically degraded by near UV and visible radiation and form oxidized groups (carbonyls and carboxyls) (Ahn et al. 2019, Egerton and Shah 1968, Pinlova and Nowack 2023).” (Lines 264-268)

Figure S15 Rayon micro-FTIR spectra comparison between samples collected from low (A-B) and high (C-D) latitudes. As

the carbonyl index is a better indicator of weathering for polymers whose initial polymer structure does not contain C=O groups, and it would not linearly increase with latitudes alteration during long-range transport, we chose to focus on rayon for a cross-latitude carbonyl index alteration analysis, which was frequently identified in samples collected. Furthermore, rayon does not contain C=O groups in pristine form. The C=O peak (at 1735 cm^{-1} , blue lines) is more clearly shown for rayon collected at higher latitudes, and the area under $1800 - 1670\text{ cm}^{-1}$ (red line intervals) was deemed as the absorbance area (A1) of the carbonyl (C=O) group, and the area under $1500 - 1390\text{ cm}^{-1}$ was deemed as the absorbance area (A2) of the reference (CH₂) group in Eq.3.

End of responses to Reviewer#2

Reviewer #3 (Remarks to the Author):

Reviewer #3 Comment 1. The paper analyses the microplastic concentrations in the atmosphere over the marine surface collected during a cruise in the Southern Ocean. The study gives important results on the field, providing with a new relevant dataset of observations in a region that was otherwise not well explored yet. While, for the relevance of the results and content of the paper I would encourage a final publication of this work on Nature Communications, I think some major revisions on the way the data, the analysis, and some of the conclusion are presented, are necessary.

Response:

We truly appreciate the positive feedback provided by the reviewer. Significant progress has been made in data analysis and adapting the interpretations, with the incorporation of comprehensive discussions that address the insightful comments and suggestions provided by all reviewers. Overall, the quality of this paper has been significantly enhanced upon revision.

Reviewer #3 Comment 2. I summarize my comments and suggestions here below:

The main weakness to the data presentation is the lack of any temporal reference, as there is no info on when the observations have been collected. It is only mentioned on the method paragraph that the campaign took place in the “2019-2020 season”. This is very vague, and it doesn’t allow for any consideration on the atmospheric processes that may have influenced the MP transport. This information is also missing from the data files provided in the attachment.

As this will be very important to understand the atmospheric conditions leading to the MP transport, and to reproduce the results of the study, it is essential that all the data also reports the representative date of collection for each sample. I do encourage putting this not only in the uploaded data, but also in the tables S1 and S1 and the figures (for example in the S4).

Response:

Thanks for this comment. Following the reviewers’ suggestion, a table summarizing the detailed sampling information has now been included in the Supplementary Information (**Table S1**).

Please find the whole table in the next page.

Table S1 Detailed information on the aerosol sampling dates and locations in the marine boundary layer and inland Antarctica. Temperature of air was obtained from the shipboard automatic weather station.

Sample No.	Sampling Period	Latitude /°N ^a	Longitude /°E ^a	Sampling Volume/m ³	Average Air Temperature/°C	Season
A01	April 20-April 22, 2020	30.20	123.06	3168	15.1	Spring
A02	April 18-April 20, 2020	24.97	128.18	3581	20.3	Spring
A03	April 16-April 18, 2020	18.67	134.05	3802	26.5	Spring
A04	April 14-April 16, 2020	12.79	139.81	3008	27.9	Spring
A05	April 12-April 14, 2020	5.12	146.94	3851	28.8	Spring
A06	April 10-April 12, 2020	-4.71	152.56	3433	29.2	Autumn
A07	April 8-April 10, 2020	-13.66	155.27	3319	28.1	Autumn
A08	April 6-April 8, 2020	-24.05	155.57	3583	25.1	Autumn
A09	April 4-April 6, 2020	-34.27	153.05	3428	20.2	Autumn
A10	April 2-April 4, 2020	-44.41	150.09	3553	13.3	Autumn
A11	November 7-November 10, 2019	-45.85	147.42	4398	7.4	Spring
A12	November 10-November 12, 2019	-54.24	144.16	3514	3.5	Spring
A13	November 12-November 14, 2019	-60.34	129.58	3467	-0.8	Spring
A14	November 14-November 16, 2019	-60.89	109.60	3506	-0.7	Spring
A15	November 16-November 18, 2019	-61.65	89.95	3486	-1.1	Spring
A16	March 14-March 16, 2020	-64.04	103.43	3566	-2.0	Autumn
A17	March 16-March 18, 2020	-64.39	128.36	3376	-2.4	Autumn
A18	March 18-March 20, 2020	-64.74	146.33	3491	-0.9	Autumn
A19	March 20-March 22, 2020	-65.17	151.61	3406	-0.7	Autumn
A20	March 22-March 24, 2020	-65.27	152.93	3506	-6.0	Autumn
A21	March 12-March 14, 2020	-65.42	82.00	3364	-6.7	Autumn
A22	November 18-November 20, 2019	-65.62	77.77	3493	-2.6	Spring
A23	November 20-November 22, 2019	-68.97	76.42	3883	-3.0	Spring
A24	November 22-November 25, 2019	-69.30	76.23	4638	-2.0	Spring
A25	January 21-January 25, 2020	-73.86	76.97	6110	-12.9	Summer
A26	January 26-January 30, 2020	-73.86	76.97	5822	-16.7	Summer

^a the mean values of individual sampling voyage legs.

Reviewer #3 Comment 3. *Also, this raises important questions on the possible interpretation of the collected data, for example: are the different samples collected at different latitudes related to different seasons? This need to be considered to understand the transport conditions and the representativeness of the data.*

Response:

Thanks very much for the comments. Yes. The different samples collected at different latitudes related to different seasons. For instance, sampling points #A12-A15 and A22-A24 were collected in November (austral spring), and sampling points A16-A21 were collected in March (austral autumn).

Although the samples were collected in different seasons, our observations in the high southern latitudes suggest that there is no significant difference for MPs and non-MPs concentrations between November (A12-A15 and A22-A24) and March (A16-

A21) (independent-samples t test; $p>0.05$). Specifically, we plotted a comparison figure between the close sampling locations within latitudes (60 °S- 65 °S) including A16-A21 (March) and A13-A15,22 (November).

Figure S4 Seasonal factors have limited influence on MPs and non-plastic particle number concentrations. A comparison between sampling locations A16-A21 (March, austral autumn) and A13-A15, 22 (November, austral spring) within latitudes (60°S- 65°S). ns: indicates no statistically significant differences between the two groups ($p>0.05$) according to t -tests.

Thus, the differences in MPs and non-plastic particles concentrations of different latitudes are unlikely to be dominated by the varied sampling seasons, and the atmospheric transport efficiency would be responsible for the spatial patterns of MPs. This point was clarified as follows. “Besides, seasonal variation was not observed in our study, and no significant difference in MPs and non-MPs concentrations between sampling locations A16-A21 (austral autumn; Table S1) and A13-A15, 22 (austral spring; Table S1) within the latitudes of 60 °S- 65 °S was observed ($p>0.05$; Figure S4).” (Lines 129-133)

Reviewer #3 Comment 4. Lines 84-85: *The authors suggest that there is a decrease in MPs fragment concentration from north to south, as if there is a linear relationship between latitude and microplastic presence in the atmosphere (hypothesis also reinforced by the use of the linear regression analysis in figure 1) which do not seem to be justified. What is this hypothesis based on?*

Response:

Thanks very much for the kind questions. In our research, continental MPs are hypothesized to be the main sources for Southern Ocean atmospheric MPs, and we used the linear regression model to fit the curves according to the suggested relationships between particle diameter and sedimentation velocity as explained below and in Text S1:

“Hicks et al. found that the dry settling velocity of atmospheric aerosol particles is directly proportional to friction velocity (Hicks et al. 2016); and Vong et al. further obtained the formula of dry settling velocity after dimensionless treatment with friction velocity (Vong et al. 2010) as follows.

$$V_d = C (U_f) D [1 + (-300/L)^{2/3}] \quad (\text{Eq S3})$$

V_d is the settling velocity and has a unit of cm s^{-1} , C is a numerical coefficient, U_f is the friction velocity and has units of m s^{-1} , L is the Monin–Obukhov length and has units of m, and D is the diameter of the particle and has units of μm .

Based on the above equation (Eq S3), we assume for a specific-sized particle, its V_d is close to a constant. Thus, we hypothesize that the relationship between distance and MPs concentrations follows a linear relationship. We also suppose that the linear decrease trend may not due to the difference in latitude, but the distance from populations and an explanation has been supplemented in the manuscript.

“Continental MPs are hypothesized to be the main source for Southern Ocean atmospheric MPs, and we used the linear regression model to fit as explained in Text S1. We used latitude as a proxy for the impacts of anthropogenic emissions, which are expected to decrease from north to south along the study transect. Thus, the linear decrease trend for fragments is not due to the difference in latitude, but rather the distance from populations. The majority of sources (i.e., land from populated urban centers in Southern Asia and Oceania) in the Southern Ocean are located to the north. However, this latitude trend may not be applicable to other regions, such as an equivalent latitude difference across a continent, because inputs from population centers across the latitudes would be continuous.” (Lines 104-113)

Reviewer #3 Comment 5. Lines 133-136: The back-trajectory analysis, which seems to carry an important message in this work (i.e. that the northern latitudes samples are related to a meaningful influence of land sources with respect to the southern ones), looks not very robust.

Response:

Thanks very much for the kind comments. As the reviewer pointed out, the back-trajectory analysis can provide an important message for the air mass directions and potential land sources. In the revised manuscript, we have included additional sampling points for back-trajectory analysis and have incorporated a more comprehensive examination of the vertical uplift of the air masses. Furthermore, we have expanded the discussion section to provide a more thorough analysis. More details are provided in the point-by-point responses to similar comments on this topic below.

Reviewer #3 Comment 6. My main concerns are the following:

- it is not clear how the trajectories release points (i.e. the samples for which the transport analysis is shown) in figure S4 are chosen. Why not putting together the air masses transport analysis for the different latitude sections instead of doing the analysis only on 6 samples?

- Panel B is cut on the northern latitudes and it is not really possible to see if the trajectories are actually reaching the land or not

Response: Thanks very much for the question. Sorry for the misunderstanding, we have previously performed all the trajectories and put six samples with different latitudes in the previous Figure S4. In the revised version, we have included back-trajectory analysis for all sampling points and have incorporated a more comprehensive examination of the vertical uplift of the air masses. In addition, panel B in the figure was re-made (A06 in the revised Figure S5).

Figure S5 Backward trajectories of all samples collected in marine boundary layer. Detailed information of samples shown in each individual panel are provided in Table S1.

Reviewer #3 Comment 7. - *The trajectory analysis does not seem to consider the vertical uplift of the air masses. That means that it is not considered if the air masses when traveling over land are reaching the surface or are above the planetary boundary layer. If the last condition is true, then the land sources may actually not have influenced the air masses and the MPs could come from the ocean instead.*

Response:

A good point! Yes, in the trajectory analysis the vertical uplift of the air masses were not initially considered. The vertical upward wind ($1-1.5 \text{ m s}^{-1}$) can occur fairly frequently in convective updrafts (e.g., Frank et al. 2013). In this case, the particles near the surface would be uplifted and released into the atmosphere. In addition, it has previously been reported that large dust particles of up to 0.45 mm in diameter can be suspended and transported to locations more than 3000 km from their source (van der Does et al. 2018). The density of dust particles is larger than those of plastic particles, and thus it is expected that the MPs can also be suspended from the surface and transported far away (Bullard et al. 2021). In the revised manuscript, we have explained this as follows.

“The transport distance of particles is intricately linked to their aero sedimentation velocity and vertical upward wind. For compact MPs fragments with effective radius, r , of approximately $60 - 80 \mu\text{m}$ and a density of $\sim 1.4 \text{ g cm}^{-3}$ (e.g., PVC), its aero sedimentation speed, U , is estimated to be between $1.3 - 1.5 \text{ m s}^{-1}$. That is, for an updraft wind with a vertical component greater than approximately 1.5 m s^{-1} , fragments of similar or smaller masses will become airborne and carried by the wind. Similarly, the lower ρ_p corresponds to lower U ; e.g., polyethylene ($0.91-0.94 \text{ g}\cdot\text{cm}^{-3}$) and polystyrene ($0.96-1.05 \text{ g}\cdot\text{cm}^{-3}$), their estimated sedimentation speed is approximately $1.0-1.2 \text{ m}\cdot\text{s}^{-1}$. As reported, the vertical upward wind ($1-1.5 \text{ m s}^{-1}$) can occur fairly frequently in convective updrafts (e.g., Frank et al. 2013). In this case, the particles near the surface would be uplifted and released into the atmosphere.” (Lines 147-157)

In terms of the air mass transport over the continents, we calculated the backward ensemble trajectory of samples that are influenced by the continental sources to show changes in elevation during air mass transport (**Figure R1**). In this case, we took four samples as an example. It is clearly shown that the air mass can be near the surface or in the planetary boundary layer ($< \sim 2000\text{m}$) when travelling over land. Therefore, we can anticipate that MPs found in our samples are mainly from the continental sourced air.

Figure R1 Backward ensemble trajectories of the samples that are influenced by the continental air mass, as well as the elevation of air mass during the transport along individual trajectories. Panels (a), (b), (c), and (d) correspond to the samples A02, A06, A07, and A08 (Table S1), respectively.

Reviewer #3 Comment 8. Also, in the description of the trajectories methods (lines 579,580) is not clear what is meant by “Back trajectories were run every 12 hours during the 7 days before the sampling date”. That sounds incorrect, as there would be no reasons to release trajectories up to 7 days before the sampling date. Is 7 days the length of the trajectories computation back in time, or the length of the release time interval? This needs to be clarified and both quantities need to be specified.

Response:

Thanks for the comments, and 7 days is the length of the trajectories computation back in time. Previously published reports suggest that the residence time for MPs particles in the atmosphere may vary between 1 and 156 h (Brahney et al. 2021), and usually the backward trajectories were run for 6-7 days to show the full range of possible sources (Aves et al. 2022). These

points were clarified in the revised version, and the detailed sample dates have been provided in the Table S1 of the revised Supporting Information.

“7-day back trajectories were run for the time of sampling (Table S1). The length of 7 days was chosen because previously published reports suggested that the residence time for MPs in the atmosphere may vary between 1 and 156 h (Brahney et al. 2021), and the backward trajectories were usually run for 6-7 days to show the full range of possible sources (Aves et al. 2022).” (Lines 714-718)

Reviewer #3 Comment 9. Lines 149-152: *This is another important concept that needs to be stated cautiously. The results from the study of Yang et al. 2022 are relative to specific sizes of fibers (~100 μm length) and it has been performed in a tank experiment. While this is a starting point to hypothesize that fibers are ejected from water with less probability with respect to spheres, it is still not enough to assume safely that the long-range transport is the responsible for the fibers' presence observed in this study. Since the fibers you observe are quite large in diameter (order of tenths μm) and length (order of hundreds μm and some even thousands), even if the morphology surely affects their lifetime in the atmosphere with respect to fragments, one can also hypothesize that are also not being suspended since too long. Are the back trajectories originating from the fibers sampling always reaching land in a reasonable time frame? And how is this related to the number of fibers observed? This can be a way to check, in case you want to assume that they are only emitted on land, which would be the time of transport from there and if it is more reasonable to hypothesize that they are actually also emitted from the sea.*

Response:

Thanks very much for the detailed questions and thoughtful thinking! For the sampling points located in the Northern Hemisphere and low latitudes (0-30 °S) Southern Hemisphere, the back trajectories originating from the sampling sites always reach land (please check the updated Figure S5); whereas for the sampling points located in the mid- and high latitudes Southern Hemisphere, their 7-d back trajectories did not reach land.

We agree with the reviewer that Yang et al. (2022) only performed a lab-scale simulation experiment with some types and lengths of fibers (PET, PVA, and PP fibers with diameters ranging from 25 – 40 μm with an average length of 100 μm), and the preliminary results cannot exclude the possibilities of MPs fibers emission from the sea. Therefore, we revised the sentences as follows.

“Our results cannot discount the possibility that MnPs are emitted from the ocean as waves break (Allen et al. 2020, Yang et al. 2022). ~~laboratory experiments indicate that fibers are not emitted from the sea surface to the atmosphere (Yang et al. 2022), hence we assume that long-range atmospheric transport is responsible.~~ While it is plausible to consider that fibers can be emitted from the ocean surface and be released to the atmosphere, in a small scale-laboratory study investigating a restricted range of polymer types and fiber lengths (diameters ranging from 25 – 40 μm with an average length of 100 μm), fibers were not emitted in significant quantities compared to spheres and fragments (Yang et al. 2022). This may be because fibers have increased surface tension compared to other MPs morphologies. Hence, we assumed that long-range atmospheric transport was responsible for most of the fibers collected in atmospheric samples in this study” (Lines 186-197)

Reviewer #3 Comment 10. Lines 168-186: *I think the analysis on the sealant tar is quite interesting, but I do wonder how the authors can exclude that this is not coming instead from ships coating (for example epoxy resins are widely used for*

this purpose) and therefore from ocean rather than the coast (or a mixture of the two).

Response:

Thank you very much for your question and suggestion of alternative sources of these particles. At first, we also considered that the origin of sealant tar might be from ships' coating because ships' coating or pigments have been deemed as emerging ocean microplastics sources. Part of the materials within pigments (i.e., polymers) are epoxy resin, acryl-styrene, and chlorinated rubber binders, etc. (Dibke et al. 2021).

We have detected the existence of epoxy resin in our samples, which composed a large component of all MPs fragments measured (13%; Figure 3B), but it is difficult to infer the specific source of resin particles as they are widely applied (Takeichi and Furukawa 2012). We think the reviewer makes a good point here, and that the sealant tar may be coming from coastal dams, but also possible from other sources such as ship coatings. Thus, we revised the manuscript as follows.

“However, they are still widely used in dams sealing worldwide (Dempster and Lannen , Liu et al.), and sometimes they may also present in ships' coating (Dibke et al. 2021).” (Lines 222-223)

Reviewer #3 Comment 11. Minor and technical comments:

Lines 61,62: In this sentence the authors seem to put together two concepts (the ocean winds/sea spray mist with the earth electric field forces) that I am not sure are really fitting with each other in the framework of that paragraph. If we want to emphasize the uncertainties related to the presence and transport of atmospheric microplastic in the general sense, at hemispheric level (as the paragraph seems to suggest), it is the identification of the sources, the mechanism of emission, resuspension and the transport and removal processes of MPs that constitute the overall uncertainty. While the electric forces may be indeed part of the general equation, the ocean winds and the sea spray represent only a small fraction of it when talking about the total sources.

Response:

Thanks very much for the thoughtful suggestions. We have rephrased the sentences and separated the two concepts as follows. The electric potential distribution on the surface of any insulating polymer is usually neglected, even though plastics are usually considered as electroneutral (de Lima et al. 2021). Polymers can present excess electrostatic charge that may persist for considerable amounts of time (Burgo et al. 2014), and these electrostatic charging can change their friction coefficients (Burgo et al. 2013).

“Additionally, the limited knowledge on the identification of MPs sources, emission mechanisms, and the transport and resuspension processes (i.e., the influence of ocean winds, sea spray mist (Allen et al. 2020, Van Sebille et al. 2016)) contribute towards the overall uncertainty. Furthermore, ~~the influence of ocean winds, sea spray mist, and the earth electric field forces, on MPs and similar sized dust particles transport likely~~ plastics can present excess electrostatic charge for considerable amounts of time (Burgo et al. 2014), which is usually neglected as they are considered as electroneutral (de Lima et al. 2021), which may increase this complexity (i.e., change their friction coefficients (Burgo et al. 2013)) in understanding the hemispheric abundance of MPs.” (Lines 63-70)

Reviewer #3 Comment 12. I guess the authors mean that, among all the uncertainties that are related to the MP transport in the atmosphere, the work is going to provide observational quantification of the presence of MP in the Southern hemisphere marine atmosphere, which will help understand better the role of the ocean as a source of MP in the bigger

picture. If that is the case, I would encourage the authors to emphasize more clearly this point.

Response:

Thanks so much for the suggestions to help us improve the clarity and meaning of our text. We agree that the emphasis of this study was not clearly stated in the previous version, and we have taken the reviewers' advice and rephrased this passage in the manuscript as follows"

"Additionally, the limited knowledge on the identification of MPs sources, emission mechanisms, and the transport and resuspension processes (i.e., the influence of ocean winds, sea spray mist (Allen et al. 2020, Van Sebille et al. 2016)) contribute towards the overall uncertainty. Furthermore, ~~the influence of ocean winds, sea spray mist, and the earth electric field forces, on MPs and similar sized dust particles transport likely~~ plastics can present excess electrostatic charge for considerable amounts of time (Burgo et al. 2014), which is usually neglected as they are considered as electroneutral (de Lima et al. 2021), which may increase this complexity (i.e., change their friction coefficients (Burgo et al. 2013)) in understanding the hemispheric abundance of MPs. Amongst the numerous uncertainties related to the transportation of MPs in the atmosphere, we present evidence of MPs in the marine atmosphere of the Southern Hemisphere, which will help in understanding the both roles of the ocean and long-distance transport as sources of MPs in remote areas." (Lines 63-74)

Reviewer #3 Comment 13. Figure 1: It is not clear to me how should I read the ** symbols over the connected histograms. Why is in panel A connecting all latitudes with Antarctica and in panel B the Northern latitudes with all the others? What does the purple horizontal bar represent?

Response:

Sorry for the misunderstanding in the figure illustration. The purple bars do not have specific meanings, but only show the connected two groups have the same statistical analysis results. To avoid confusion, the purple bars (in Figure 1) have been changed to the normal black lines.

We have revised the expression ways for the statistical analysis. The groups under black lines showed the same statistical analysis results with the comparing groups. In this way, we suppose the meaning that fibers concentrations were relatively constant, and the fragments concentrations were decreasing along the altitudes can be better explained.

Figure 1 Number concentrations of MPs along the cruise path from the mid-Northern Hemisphere to Antarctica. **(A)** Fibers and **(B)** Fragments concentrations at low, mid, and high latitudes along the cruise path and over inland Antarctica; **(C)** MPs fibers, **(D)** MPs fragments, **(E)** non-plastic fibers, and **(F)** non-plastic fragments variations in concentration along the cruise path. *t*-test analysis between the inland Antarctic sites and other latitude regions were performed in panels A and B, and a stepwise linear regression was used to determine the relationship between particles concentrations and latitudes in panels C-F. Symbols of *, **, and *** indicates *p* value < 0.05, 0.01, and 0.001, respectively. The groups under black lines showed the same statistical analysis results with the comparing groups.

Reviewer #3 Comment 14. Lines 144,145: This sentence is not clear, there may be a typo. Do you mean “Similar phenomena OF constant concentrations...”?

Response:

Thanks very much for pointing out our mistake here. We have revised the sentence as follows:

“Similar phenomena of ~~that~~ constant concentrations of non-plastic fibers and decreasing burdens of non-plastic fragments were also observed along the cruise path.” (Lines 180-182)

Reviewer #3 Comment 15. Lines 184-186: Do you mean here that the dominance of white-transparent colors for fibers indicates they were subjected to longer aging (hence long-range transport) with respect to the colored fragment? It is not clear how you arrive to your conclusion, it will be useful if you could elaborate more explicitly on this point.

Response:

Thanks for pointing this out. We previously supposed that the dominance of white-transparent colors of fibers can indicate they were subjected to longer aging. However, the yellowing effects on MPs fibers were not obvious, which is a characteristic feature of MPs aging. Thus, we think this statement is not supported and have removed this statement in the revised manuscript.

~~“This phenomenon seems to be consistent with the high transport efficiency of fibers along the transect and the relatively low transport capacity of the fragments.” (Lines 230-231)~~

Reviewer #3 Comment 16. Line 399: I would suggest to explicitly say what the ANOVA analysis is, at least the first time you mention it.

Response: Thanks so much for the detailed help. We have corrected this point by explicitly showing what ANOVA analysis is.

“Letters above violin plots correspond to statistical comparison by ~~ANOVA~~ one-way analysis of variance (ANOVA) tests.” (Lines 494-495)

End of responses to Reviewer#3

References used in supporting the responses to reviewers:

- Ahn, K., Zaccaron, S., Zwirchmayr, N.S., Hettegger, H., Hofinger, A., Bacher, M., Henniges, U., Hosoya, T., Potthast, A. and Rosenau, T. (2019) Yellowing and brightness reversion of celluloses: CO or COOH, who is the culprit? *Cellulose* 26, 429-444.
- Allen, S., Allen, D., Baladima, F., Phoenix, V., Thomas, J., Le Roux, G. and Sonke, J. (2021) Evidence of free tropospheric and long-range transport of microplastic at Pic du Midi Observatory. *Nat. Commun.* 12(1), 1-10.
- Allen, S., Allen, D., Moss, K., Le Roux, G., Phoenix, V.R. and Sonke, J.E. (2020) Examination of the ocean as a source for atmospheric microplastics. *PLoS one* 15(5), e0232746.
- Aves, A.R., Revell, L.E., Gaw, S., Ruffell, H., Schuddeboom, A., Wotherspoon, N.E., LaRue, M. and McDonald, A.J. (2022) First evidence of microplastics in Antarctic snow. *Cryosphere* 16(6), 2127-2145.
- Brahney, J., Mahowald, N., Prank, M., Cornwell, G., Klimont, Z., Matsui, H. and Prather, K.A. (2021) Constraining the atmospheric limb of the plastic cycle. *Proc. Natl. Acad. Sci.* 118(16), e2020719118.
- Bullard, J.E., Ockelford, A., O'Brien, P. and McKenna Neuman, C. (2021) Preferential transport of microplastics by wind. *Atmos. Environ.* 245, 118038.
- Burgo, T.A., Silva, C.A., Balestrin, L.B. and Galembeck, F. (2013) Friction coefficient dependence on electrostatic tribocharging. *Sci-Rep UK* 3(1), 2384.
- Burgo, T.A.L., Balestrin, L.B.S. and Galembeck, F. (2014) Corona charging and potential decay on oxidized polyethylene surfaces. *Polym. Degrad. Stabil.* 104, 11-17.
- Cai, H., Du, F., Li, L., Li, B., Li, J. and Shi, H. (2019) A practical approach based on FT-IR spectroscopy for identification of semi-synthetic and natural celluloses in microplastic investigation. *Sci. Total Environ.* 669, 692-701.
- Comnea-Stancu, I.R., Wieland, K., Ramer, G., Schwaighofer, A. and Lendl, B. (2017) On the identification of rayon/viscose as a major fraction of microplastics in the marine environment: discrimination between natural and manmade cellulosic fibers using Fourier transform infrared spectroscopy. *Appl. spectrosc.* 71(5), 939-950.
- de Lima, B., Augusto, T., Rezende, C.A., Bertazzo, S., Galembeck, A. and Galembeck, F. (2021) Electric potential decay on polyethylene: role of atmospheric water on electric charge build-up and dissipation. *J. Electrostat.* 69(4), 401-409.
- Dempster, K. and Lannen, N. *Breaclaich Dam—upstream face joint bandage sealant and wavewall refurbishment works*, pp. 456-468, Thomas Telford Publishing.
- Dibke, C., Fischer, M. and Scholz-Böttcher, B.M. (2021) Microplastic mass concentrations and distribution in German bight waters by pyrolysis–gas chromatography–mass spectrometry/thermochemolysis reveal potential impact of marine coatings: do ships leave skid marks? *Environ. Sci. Technol.* 55(4), 2285-2295.
- Egerton, G.S. and Shah, K.M. (1968) The Effect of temperature on the photochemical degradation of textile materials: Part I: Degradation sensitized by titanium dioxide. *Text. Res. J.* 38(2), 130-135.
- Frank, J.M., Massman, W.J. and Ewers, B.E. (2013) Underestimates of sensible heat flux due to vertical velocity measurement errors in non-orthogonal sonic anemometers. *Agr. Forest Meteorol.* 171-172, 72-81.
- González-Pleiter, M., Edo, C., Aguilera, Á., Viúdez-Moreiras, D., Pulido-Reyes, G., González-Toril, E., Osuna, S., de Diego-Castilla, G., Leganés, F. and Fernández-Piñas, F. (2021) Occurrence and transport of microplastics sampled within and above the planetary boundary layer. *Sci. Total Environ.* 761, 143213.
- Hicks, B.B., Saylor, R.D. and Baker, B.D. (2016) Dry deposition of particles to canopies—A look back and the road forward. *J. Geophys. Res. Atmos.* 121(24), 14-691.
- Kernchen, S., Löder, M.G., Fischer, F., Fischer, D., Moses, S.R., Georgi, C., Nölscher, A.C., Held, A. and Laforsch, C. (2022) Airborne microplastic concentrations and deposition across the Weser River catchment. *Sci. Total Environ.* 818, 151812.
- Koelmans, A.A., Redondo-Hasselerharm, P.E., Mohamed Nor, N.H. and Kooi, M. (2020) Solving the nonalignment of methods and approaches used in microplastic research to consistently characterize risk. *Environ. Sci. Technol.* 54(19), 12307-12315.
- Li, J., Wang, L., Xu, Z., Zhang, J., Li, J., Lu, X., Yan, R. and Tang, Y. (2023) A new point to correlate the multi-dimensional assessment for the aging process of microfibers. *Water Res.* 235, 119933.
- Liu, K., Wang, X., Fang, T., Xu, P., Zhu, L. and Li, D. (2019a) Source and potential risk assessment of suspended atmospheric microplastics in Shanghai. *Sci. Total Environ.* 675, 462-471.
- Liu, K., Wu, T., Wang, X., Song, Z., Zong, C., Wei, N. and Li, D. (2019b) Consistent transport of terrestrial microplastics to the ocean through atmosphere. *Environ. Sci. Technol.* 53(18), 10612-10619.
- Liu, Z., Jia, J., Zhao, C., Zheng, C., Qiu, Z. and Li, A. Application of underwater repair technology for dams in China. 2020 International Conference on Ecological Resources, Energy, Construction, Transportation and Materials (EECTM 2020).
- Mintenig, S.M., Kooi, M., Erich, M.W., Primpke, S., Redondo-Hasselerharm, P.E., Dekker, S.C., Koelmans, A.A. and Van Wezel, A.P. (2020) A systems approach to understand microplastic occurrence and variability in Dutch

- riverine surface waters. *Water Res.* 176, 115723.
- Moch, J.M., Dovrou, E., Mickley, L.J., Keutsch, F.N., Liu, Z., Wang, Y., Dombek, T.L., Kuwata, M., Budisulistiorini, S.H. and Yang, L. (2020) Global importance of hydroxymethanesulfonate in ambient particulate matter: Implications for air quality. *J. Geophys. Res-Atmos.* 125(18), e2020JD032706.
- Pinlova, B. and Nowack, B. (2023) Characterization of fiber fragments released from polyester textiles during UV weathering. *Environ. Pollut.* 322, 121012.
- Simon, M., van Alst, N. and Vollertsen, J. (2018) Quantification of microplastic mass and removal rates at wastewater treatment plants applying Focal Plane Array (FPA)-based Fourier Transform Infrared (FT-IR) imaging. *Water Res.* 142, 1-9.
- Takeichi, T. and Furukawa, N. (2012) Epoxy resins and phenol-formaldehyde resins. *Polymer Science: A Comprehensive Reference*.
- Trainic, M., Flores, J.M., Pinkas, I., Pedrotti, M.L., Lombard, F., Bourdin, G., Gorsky, G., Boss, E., Rudich, Y., Vardi, A. and Koren, I. (2020) Airborne microplastic particles detected in the remote marine atmosphere. *Commun. Earth Environ.* 1(1), 64.
- Uddin, S., Fowler, S.W., Habibi, N., Sajid, S., Dupont, S. and Behbehani, M. (2022) A preliminary assessment of size-fractionated microplastics in indoor aerosol—Kuwait’s baseline. *Toxics* 10(2), 71.
- van der Does, M., Knippertz, P., Zschenderlein, P., Giles Harrison, R. and Stuut, J.-B.W. (2018) The mysterious long-range transport of giant mineral dust particles. *Sci. Adv.* 4(12), eaau2768.
- Van Sebille, E., Spathi, C. and Gilbert, A. (2016) The ocean plastic pollution challenge: towards solutions in the UK. *Grant. Brief. Pap.* 19, 1-16.
- Vong, R.J., Vong, I.J., Vickers, D. and Covert, D.S. (2010) Size-dependent aerosol deposition velocities during BEARPEX'07. *Atmospheric Chem. Phys.* 10(12), 5749-5758.
- Wang, X., Liu, K., Zhu, L., Li, C., Song, Z. and Li, D. (2021) Efficient transport of atmospheric microplastics onto the continent via the East Asian summer monsoon. *J. Hazard. Mater.* 414, 125477.
- Yang, S., Zhang, T., Gan, Y., Lu, X., Chen, H., Chen, J., Yang, X. and Wang, X. (2022) Constraining microplastic particle emission flux from the ocean. *Environ. Sci. Technol. Lett.* 9(6), 513-519.
- Zhang, K., Su, J., Xiong, X., Wu, X., Wu, C. and Liu, J. (2020) Microplastics differ between indoor and outdoor air masses: insights from multiple microscopy methodologies. *Appl. spectrosc.* 74(9), 1079-1098.

Reviewer #2 (Remarks to the Author):

I do not have any further comments and I am happy with the revisions

Reviewer #3 (Remarks to the Author):

The work is bringing new and relevant knowledge about the problem of atmospheric microplastic and its possible sources. Since the reviewed paper has noticeably improved, and the research results are now presented in a more robust and sound way, I do support the publication of the reviewed manuscript without further revisions.

Response to Reviewers' Comments

Reviewer #2 (Remarks to the Author):

I do not have any further comments and I am happy with the revisions

Reviewer #3 (Remarks to the Author):

The work is bringing new and relevant knowledge about the problem of atmospheric microplastic and its possible sources. Since the reviewed paper has noticeably improved, and the research results are now presented in a more robust and sound way, I do support the publication of the reviewed manuscript without further revisions.

Response:

We thank the both reviewers very much for their time in evaluating our work. Also, we appreciate their positive comments on our revised version.